# Disordered breathing in a Pitt-Hopkins syndrome model involves Phox2b-expressing parafacial neurons and aberrant Nav1.8 expression

C. M. Cleary [1], S. James[1], B. J. Maher[2,3,4] & D. K. Mulkey [1]✉

Pitt-Hopkins syndrome (PTHS) is a rare autism spectrum-like disorder characterized by intellectual disability, developmental delays, and breathing problems involving episodes of hyperventilation followed by apnea. PTHS is caused by functional haploinsufficiency of the gene encoding transcription factor 4 (*Tcf4*). Despite the severity of this disease, mechanisms contributing to PTHS behavioral abnormalities are not well understood. Here, we show that a *Tcf4* truncation (*Tcf4*$^{tr/+}$) mouse model of PTHS exhibits breathing problems similar to PTHS patients. This behavioral deficit is associated with selective loss of putative expiratory par- afacial neurons and compromised function of neurons in the retrotrapezoid nucleus that regulate breathing in response to tissue $CO_2/H^+$. We also show that central Nav1.8 channels can be targeted pharmacologically to improve respiratory function at the cellular and behavioral levels in *Tcf4*$^{tr/+}$ mice, thus establishing Nav1.8 as a high priority target with therapeutic potential in PTHS.

[1] Department of Physiology and Neurobiology, University of Connecticut, Storrs, CT, USA. [2] Lieber Institute for Brain Development, Johns Hopkins Medical Campus, Baltimore, MD 21205, USA. [3] Department of Psychiatry and Behavioral Sciences, Johns Hopkins School of Medicine, Baltimore, MD 21287, USA. [4] Department of Neuroscience, Johns Hopkins School of Medicine, Baltimore, MD 21205, USA. ✉email: daniel.mulkey@uconn.edu

itt-Hopkins syndrome (PTHS) is an autism spectrum dis-
order caused by haploinsufficiency of the gene encoding
transcription factor 4 (*Tcf4*; GeneID: 6925)[1]. The symptoms
of PTHS include intellectual disability, developmental delay, sei-
zures, and disordered breathing during wakefulness characterized
by episodes of hyperventilation with intermittent apnea or breath
hold[2,3]. Breathing problems associated with this disease have a
negative impact on quality of life[4] and likely contribute to
aspiration-induced pneumonia, which is the leading cause of
death in PTHS[5]. Despite this, virtually nothing is known
regarding how *Tcf4* deficiency disrupts respiratory control; con-
sequently, candidate therapeutic targets are lacking. Interestingly,
recent evidence suggests that loss of *Tcf4* causes aberrant
expression of *Scn10a* in the brain[6]; however, the therapeutic
potential of this target remains largely untested.

Main elements of respiratory control include central chemor-
eceptors which regulate breathing in response to changes in tissue
$CO_2/pH$[7], the pre-Bötzinger complex (pre-BötC) which generates
inspiratory rhythm[8], and respiratory motor neurons that serve as
the final common pathway to respiratory muscle[9]. Although loss
of *Tcf4* may disrupt breathing at any level of the respiratory
circuit, clinical evidence suggests disruption of central chemor-
eception is a contributing factor. For example, disordered
breathing in PTHS is phenotypically similar to a related disorder
known as Rett syndrome (RTT)[10], and breathing problems in
RTT involve disruption of central chemoreception[11,12]. Also
consistent with this possibility, acetazolamide—a carbonic
anhydrase inhibitor used to induce metabolic acidosis and
hyperventilation[13]—improved breathing in PTHS patients[14,15].
The retrotrapezoid nucleus (RTN) is an important respiratory
control center located in the ventral parafacial region of the
medulla[16]. A subset of glutamatergic neurons in this region
function as respiratory chemoreceptors by sensing changes in
tissue $CO_2/H^+$ and communicating this information to other
elements of the respiratory circuit, including the pre-BötC, to
regulate inspiratory activity[17]. RTN chemoreceptors also project
to expiratory centers to elicit active expiration during high $CO_2$[18];
however, recent evidence suggests this function is controlled by a
group of glutamatergic neurons located in the adjacent parafacial
lateral region ($pF_L$)[19]. Transcription factors essential for normal
development and function of RTN chemoreceptors include
paired-like homeobox 2b (*Phox2b*)[20,21] and atonal homolog 1
(*Atoh1*; a.k.a. *Math1*)[21–23]. Evidence also suggests that *Atoh1*
forms a functional complex with *Tcf4* to regulate brainstem
development in a cell-type specific manner[24]. Based on this, we
wondered whether loss of *Tcf4* disrupts development and func-
tion of RTN chemoreceptors or $pF_L$ neurons and contributes to
disordered breathing in PTHS.

Here, we provide the first characterization of disordered
breathing in a mouse model of PTHS. We show that under room
air conditions *Tcf4*$^{tr/+}$ mice show frequent episodes of hyper-
ventilation as well as reduced sigh activity and increased post-sigh
apnea. These mice also fail to increase inspiratory and expiratory
output in response to $CO_2$. The basis for this behavior deficit
involves selective loss of parafacial Phox2b+ neurons, altered
connectivity between Phox2b+ parafacial neurons and the pre-
BötC, and suppressed excitability of chemosensitive RTN neu-
rons. We also show that central Nav1.8 channels can be targeted
pharmacologically to improve chemoreceptor function in *Tcf4*$^{tr/+}$
mice, establishing Nav1.8 as a high priority target with ther-
apeutic potential in PTHS.

## Results

*Tcf4*$^{tr/+}$ mice (JAX stock # 013598) were crossed with each other
or *Tcf4*$^{+/+}$ control mice (common 50:50 background of 129S1/

SvlmJ::C57BL6/J) (Supplementary Fig. 1); control and hetero-
zygous mice were obtained at the expected frequencies at birth.
However, consistent with evidence that homozygous loss of *Tcf4*
results in embryonic lethality[6,25], we found that ~30% *Tcf4*$^{tr/tr}$
mice were stillborn. Those *Tcf4*$^{tr/tr}$ pups born alive had a reduced
body weight compared to control ($F_{2,27} = 56.86$, $p < 0.0001$) and
died within the first few days of life, reaching 100% mortality by
4 days of age (Fig. 1A, B). At weaning age, *Tcf4*$^{tr/+}$ mice also
show reduced body weight ($T_{28} = 3.378$, $p = 0.0022$) and
increased mortality (30%) compared to *Tcf4*$^{+/+}$ mice (0%)
(Fig. 1A, B); however, by adulthood (44 days of postnatal), the
size of *Tcf4*$^{tr/+}$ mice was similar to *Tcf4*$^{+/+}$ ($T_{28} = 0.6590$,
$p > 0.05$). Despite their normal size, adult *Tcf4*$^{tr/+}$ mice are
known to exhibit a variety of abnormal behaviors including
hyperactivity[26,27] and decreased anxiety[26]. Therefore, we char-
acterized these behaviors using the novel open field assay and
found that *Tcf4*$^{tr/+}$ mice show more locomotor activity (total
distance traveled) and higher frequency of entering the center
region (inversely related to anxiety) compared to littermate
control mice (Fig. 1C–E). These results confirm that *Tcf4*$^{tr/+}$ mice
used in this study exhibit behavior abnormalities consistent with
similar PTHS models[26–28]. A subset of PTHS patients also exhibit
seizure activity[29]. Therefore, we used radio telemetry to record
electrocorticogram (ECoG) and electromyography (EMG) activ-
ity over a 24-h period in *Tcf4*$^{tr/+}$ and control mice. We found that
*Tcf4*$^{tr/+}$ mice (60–64 days postnatal; $n = 6$ mixed sex) did not
exhibit overt seizures or seizure-like ECoG activity (large ampli-
tude poly-spike activity) for the duration our recording (Sup-
plementary Fig. 2). These results suggest *Tcf4*$^{tr/+}$ mice at this
developmental time-point do not show spontaneous seizure-like
activity.

To determine whether *Tcf4*$^{tr/+}$ mice exhibit breathing problems
similar to PTHS patients, we used whole-body plethysmography to
measure respiratory activity in awake mice (~45 days old) under
room air conditions and during exposure to graded increases in
$CO_2$ (balance $O_2$ to minimize input from peripheral chemor-
eceptors). We found in room air that average minute ventilation
was similar between genotypes (see below). Consistent with this,
arterial $CO_2/H^+$ levels were similar in control and *Tcf4*$^{tr/+}$ mice
and both genotypes showed comparable levels of metabolic activity
across the light/dark 24-h cycle (Supplementary Fig. 3). We also
analyzed an expanded section of data recorded under room air
conditions (20 min without behavioral artifact), and found that
*Tcf4*$^{tr/+}$ mice ($n = 6$, mixed sex) show periodic breathing char-
acterized by repeated cycles of waxing and waning of minute ven-
tilation. This pattern of activity was not observed in *Tcf4*$^{+/+}$ mice
($n = 6$, mixed sex; Fig. 2A, B) and resulted in unstable breathing as
evidenced by a large increase in minute ventilation coefficient of
variation (0.09 *Tcf4*$^{t+/+}$ vs. 0.35 *Tcf4*$^{tr/+}$; $T_{10} = 12.72$, $p < 0.0001$;
Fig. 2A–C). These results show that *Tcf4*$^{tr/+}$ exhibit a respiratory
phenotype similar to PTHS patients. We also found under room air
conditions that ~75% of adult *Tcf4*$^{tr/+}$ mice ($n = 12$ mice per
genotype) exhibited a diminished occurrence of spontaneous sighs
($T_{10} = 2.774$, $p = 0.0197$) in conjunction with increased duration of
post-sigh apnea ($T_9 = 2.490$, $p = 0.0344$) compared to littermate
controls (Fig. 2D–F). The occurrence of spontaneous apneic events
was similar between genotypes ($T_8 = 1.078$, $p > 0.05$). The pre-
valence of breathing problems in *Tcf4*$^{tr/+}$ mice is similar to that
described for PTHS patients (over 50%)[3], and both species exhibit
frequent episodes of hyperventilation and apnea (Fig. 2A–F).

To determine whether chemoreceptor function is disrupted in
*Tcf4*$^{tr/+}$ mice, we characterized their ventilatory response to $CO_2$.
We found that *Tcf4*$^{tr/+}$ mice show a diminished capacity to
increase minute ventilation—the product of frequency and tidal
volume—in response to 5 and 7% $CO_2$ ($m = 1.145$ control vs.
$m = 0.5453$ *Tcf4*$^{tr/+}$; $F_{1,4} = 22.22$, $p = 0.0092$) (Fig. 2G, H). This

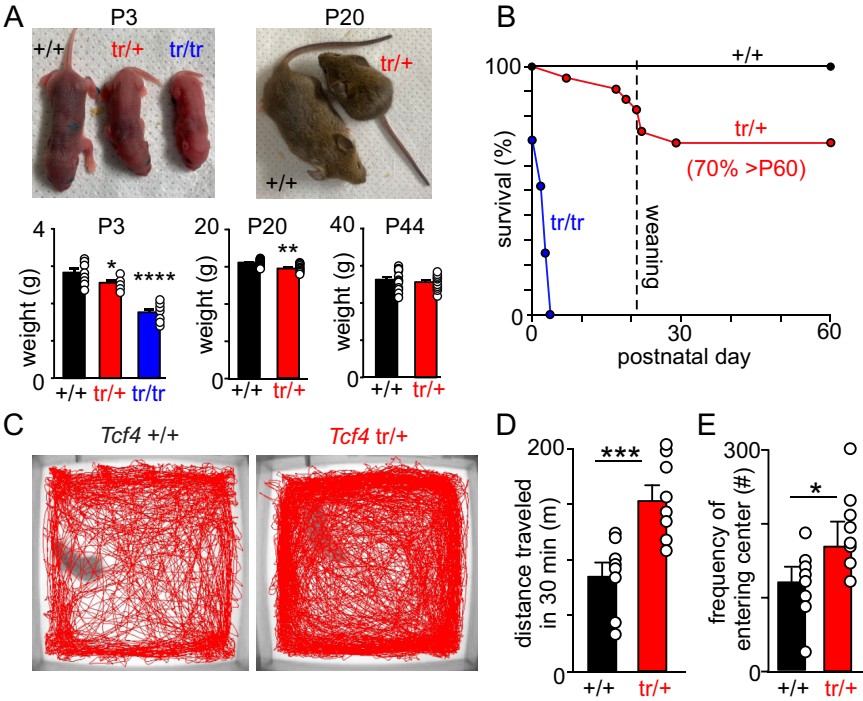

**Fig. 1 Survival and locomotor abnormalities exhibited by *Tcf4*tr/+ mice. A** Mouse images and summary data show that *Tcf4*tr/+ and *Tcf4*tr/tr mice are smaller and weigh less early in development compared to *Tcf4*+/+ control mice (day 3: $F_{2,27} = 56.86$, $p < 0.0001$; day 20: $T_{28} = 3.378$, $p = 0.0022$, data are presented as mean values ± SEM); however, by 44 days of age *Tcf4*tr/+ and *Tcf4*+/+ are of similar size ($n = 30$ animals/15 each genotype, $T_{28} = 0.6590$, $p > 0.05$, paired *t*-test, data are presented as mean values ± SEM). **B** Survival curves show that ~30% of *Tcf4*tr/tr mice are born dead and reach 100% lethality by 4 days of age. *Tcf4*tr/+ also exhibit early high mortality but those reaching weaning age tended to survive at least two months. **C** Representative locomotor activity maps of *Tcf4*+/+ and *Tcf4*tr/+ mice (40–50 days of age) during a 30-min periods following placement in a novel open field arena. **D**, **E** summary data show that *Tcf4*tr/+ traveled further (**D** $n = 15$ biologically independent animals, mixed sex, $T_{14} = 4.008$, $p = 0.0013$, paired *t*-test, data are presented as mean values ± SEM) and more frequently entered the center region (middle 50% of total area) (**E** $n = 15$ biologically independent animals, mixed sex, $T_{14} = 2.559$, $p = 0.0227$, paired *t*-test, data are presented as mean values ± SEM) compared to *Tcf4*+/+. Asterisk (*) indicate different between genotypes (unpaired *t*-test). One symbol = $p < 0.05$, two symbols = $p < 0.01$, three symbols = $p < 0.001$, four symbols = $p < 0.0001$.

respiratory phenotype is specific to central chemoreception since *Tcf4*tr/+ adult mice and pups showed a normal ventilatory response to hypoxia (10% $O_2$; balance $N_2$, $F_{1,3} = 0.036$, $p > 0.05$) (Supplementary Fig. 4). It should be noted that the above experiments were performed using mice housed in a normal 12:12 light/dark cycle, and experiments were conducted during the light/inactive state. However, since breathing problems in PTHS occur primarily during wakefulness[3], we also characterized respiratory activity during the dark/active state in mice housed under reverse light-dark cycle conditions. We found that *Tcf4*tr/+ mice exhibit a similar respiratory phenotype under both light and dark cycle conditions. Specifically, during the dark/active state *Tcf4*tr/+ mice ($n = 6$; 30–40 days of age, mixed sex) showed reduced sigh frequency and increase duration of post-sigh apneas under baseline conditions, and a blunted ventilatory response to $CO_2$. These results suggest that disordered breathing in *Tcf4*tr/+ mice during wakefulness is similar between the dark/active and light/inactive periods.

An important aspect of the $CO_2$ ventilatory response is recruitment of active expiration[7], and surprisingly, *Tcf4*tr/+ completely lack this feature of the chemoreflex. For example, abdominal (expiratory activity) and diaphragm (inspiratory activity) electromyogram (EMG) activity was measured in isoflurane (1.5%) anesthetized *Tcf4*+/+ and *Tcf4*tr/+ mice during exposure to high $CO_2$. As expected, *Tcf4*+/+ mice showed a characteristic dose-dependent increase in abdominal EMG activity in response to 5 and 7% $CO_2$ (Fig. 3A, D, E). Conversely, *Tcf4*tr/+ mice showed no abdominal EMG response up to 7% $CO_2$ (Fig. 3A, D, E). Also, consistent with observations made in awake

mice, anesthetized *Tcf4*tr/+ showed normal respiratory activity in room air but failed to increase diaphragm EMG amplitude ($F_{1,5} = 72.89$, $p = 0.0004$) and frequency ($F_{1,5} = 52.07$, $p = 0.0008$) during exposure to 5 and 7% $CO_2$ (Fig. 3A–C). Together, these results show that both inspiratory and expiratory responses to $CO_2$ are disrupted in *Tcf4*tr/+ mice.

**Cellular basis of disordered breathing in PTHS**. Considering *Tcf4* is required for differentiation of subsets of *Atoh1*+ progenitors[24] and since *Atoh1* is required for development of *Phox2b*-expressing parafacial neurons[22,23], we wanted to determine if these transcription factors are co-expressed by Phox2b+ neurons in this region, and whether the RTN or pF_L are disrupted by loss of *Tcf4*. To address the first possibility, we performed single cell RNA sequencing (scRNA-seq) on cells isolated from the ventral parafacial region of *Tcf4*+/+ pups (9–11 days of age). We confirm that RTN chemoreceptors are comprised of two subsets (clusters 1–2) of glutamatergic *Phox2b*-expressing and *Nmb*-expressing neurons with similar levels of proton sensing machinery (*Gpr4* and *Kcnk5*) but differ in expression of galanin (*Gal*) and $Ca^{2+}$-dependent secretion activator 2 (*Cadps2*) (Fig. 4A). The molecular profile of expiratory pF_L neurons is less well defined, but previous evidence suggests these cells are glutamatergic and Phox2b-negataive[30]. Sympathetic C1 catecholamine neurons (cluster 3) are identified by expression of tyrosine hydroxylase (*Th*) and *Phox2b* and the absence of *Nmb*[31]. We found that *Tcf4* and *Atoh1* are co-expressed in clusters 1–2, which match the molecular signature of RTN chemoreceptors (Fig. 4A).

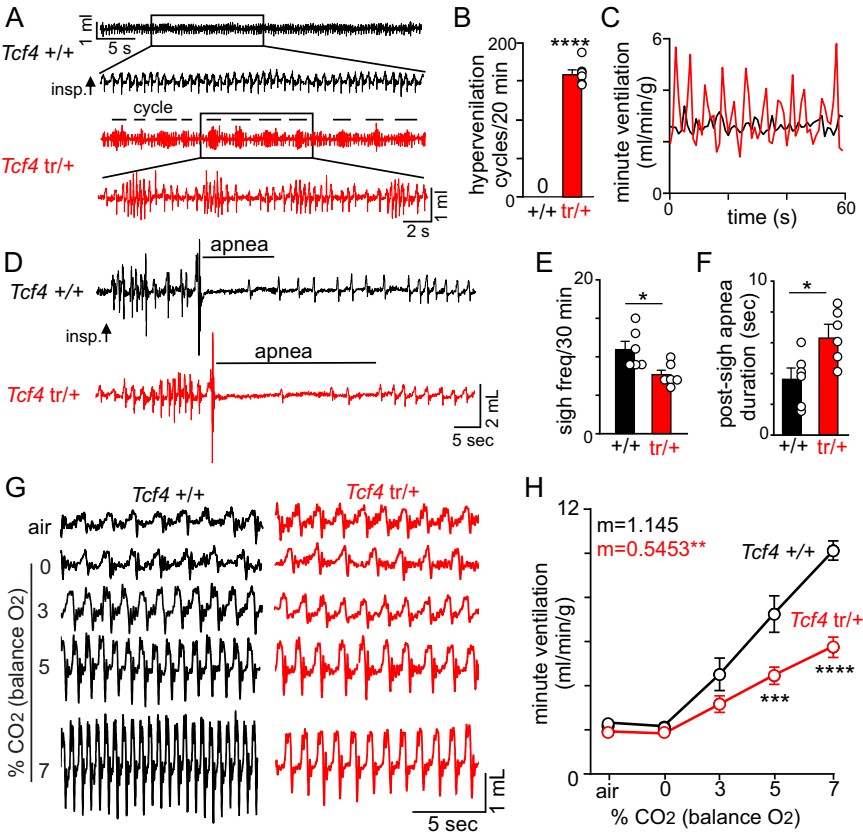

**Fig. 2 $Tcf4^{tr/+}$ mice exhibit unstable breathing under baseline conditions and a blunted ventilatory response to $CO_2$. A** Traces of respiratory activity show that under room air conditions $Tcf4^{tr/+}$ mice exhibit frequent cycles (designated by a horizontal line) of hyperventilation followed by a period of reduced respiratory activity. Note that this pattern of activity was not observed in $Tcf4^{+/+}$ mice. **B** summary plot showing the number of hyperventilation cycles that occurred over 20 min; $Tcf4^{tr/+}$ mice ranged from 5.9 to 9.1 cycles/min. ($N = 6$ biologically independent animals/genotype, $T_5 = 25.31$, $p < 0.0001$, one sample $t$-test, data are presented as mean values ± SEM). **C** Traces of minute ventilation (same animals as **A**) show the unstable periodic nature of respiratory activity exhibited by $Tcf4^{tr/+}$ mice compared to control. **D** Traces of respiratory activity under room air conditions from a $Tcf4^{+/+}$ and $Tcf4^{tr/+}$ mouse show examples of post-sigh apnea in each genotype. **E**, **F** Summary data show that under room air conditions $Tcf4^{tr/+}$ mice exhibited less frequent sighs (**E** $n = 6$ biologically independent animals/genotype, $T_{10} = 2.774$, $p = 0.0197$, data are presented as mean values ± SEM) and longer duration post-sigh apnea (**F** $n = 6$ biologically independent animals/genotype, $T_{10} = 2.490$, $p = 0.0344$, data are presented as mean values ± SEM) compared to $Tcf4^{+/+}$ mice. **G** Traces of respiratory activity from a $Tcf4^{+/+}$ and $Tcf4^{tr/+}$ mouse in room air and during exposure to 0–7% $CO_2$ (balance $O_2$). **H** Plot of minute ventilation shows that $Tcf4^{tr/+}$ mice have a severely blunted $CO_2/H^+$ response compared to control mice ($n = 5$ biological independent animals/genotype, $F_{1,4} = 22.22$, $p = 0.0092$, data are presented as mean values ± SEM). Asterisk (*) indicate the different between genotypes ($t$-test or two-way ANOVA followed by Tukey's multiple comparison test). Linear regressions were compared by two-tailed analysis of covariance (ANCOVA). One symbol = $p < 0.05$, two symbols = $p < 0.01$, three symbols = $p < 0.001$, four symbols = $p < 0.0001$.

Furthermore, fluorescent in situ hybridization using tissue from $Tcf4^{+/+}$ (12 days of postnatal) shows that 89% of $Tcf4$ labeling in RTN neurons co-localized with $Atoh1$ (Fig. 4B). The remaining 11% of $Tcf4$-labled RTN neurons lacked $Atoh1$ signal (Fig. 4B). We also obtained an enriched population of Phox2b+ parafacial neurons (from 22 day old $Phox2b^{Cre}$::TdT+ mice) and subsequent qPCR confirmed the expression of both $Tcf4$ (average raw Ct value: $26.9 \pm 0.1$) and $Atoh1$ (average raw Ct value: $30.8 \pm 0.2$) transcript. Consistent with evidence from an earlier developmental time point[20], $Atoh1$ was not detected in C1 neurons (cluster 3) (Fig. 4A). These results show that $Tcf4$ and $Atoh1$ are expressed by Phox2b+ parafacial neurons, and thus are in position to coordinate development of these cells.

Consistent with this possibility, we found fewer Phox2b-immunoreactive neurons along the rostrocaudal extent of the RTN and $pF_L$ regions from $Tcf4^{tr/+}$ mice compared to control tissue (Fig. 4C). In particular, we found 70% fewer Phox2b labeled neurons in the $pF_L$ region from $Tcf4^{tr/+}$ mice compared to control. The RTN also showed a 21% loss of Phox2b-immunoreactivity and those remaining neurons tended to cluster near the ventral surface in the

medial RTN region (Fig. 4C, D). These anatomical deficits are present at birth and were similar in $Tcf4^{tr/+}$ and $Tcf4^{tr/tr}$ pups (Supplementary Fig. 5), indicating both alleles of $Tcf4$ are required early in development. Furthermore, although a subset of RTN astrocytes are derived from Phox2b-expressing progenitor cells[32], parafacial astrocytes in $Tcf4^{tr/+}$ mice crossed with an astrocyte specific inducible reporter ($Gfap^{Cre/ERT2}$) did not express $Tcf4$ transcript ($n = 3$ animals/genotype, three technical replicates per sample, Taqman probe: Mm00443210_m1) and appear morphologically normal, with no difference in abundance between genotypes ($T_4 = 1.126$, $p > 0.05$). Looking at the larger population of glutamatergic parafacial neurons (identified by expression of $Slc17a6$, the gene encoding vesicular glutamate transporter 2), the proportion of $Slc17a6+$ and $Phox2b+$ neurons decreased from 61 to 47%, while the proportion of $Slc17a6+$ and $Phox2b$-negative neurons was similar between genotypes (Supplementary Fig. 6). Looking at the larger population of glutamatergic parafacial neurons (identified by expression of $Slc17a6$, the gene encoding vesicular glutamate transporter 2), the proportion that are $Slc17a6+$ and $Phox2b+$ decreased from 61 to 47%, while the proportion of $Slc17a6+$ and

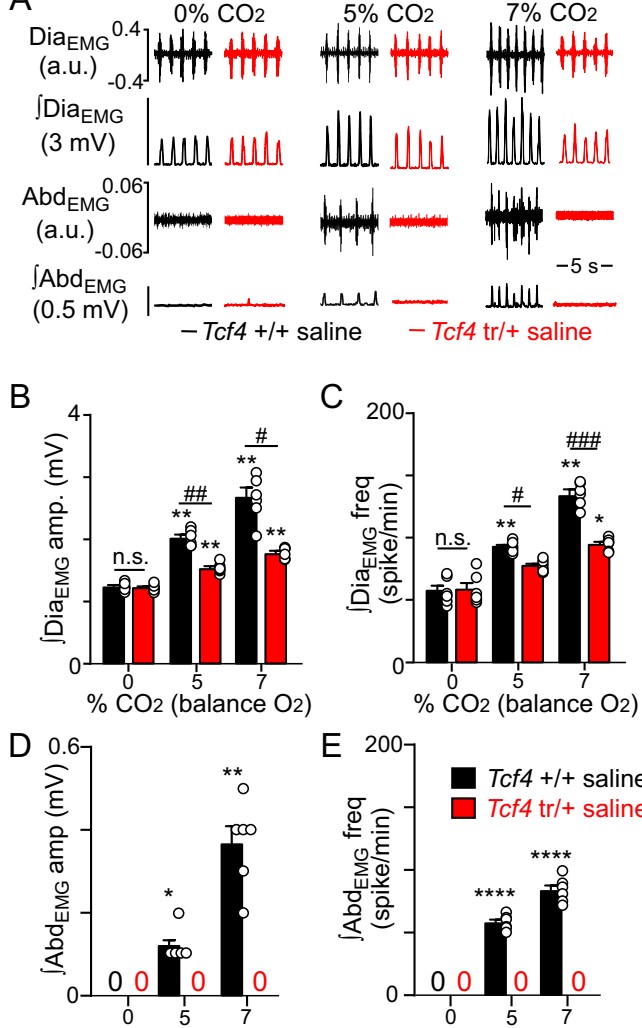

**Fig. 3 Hypercapnia fails to stimulate active expiration in anesthetized** $Tcf4^{tr/+}$ **mice.** Diaphragm and abdominal EMG activity was measured in isoflurane (1.5%) anesthetized $Tcf4^{+/+}$ and $Tcf4^{tr/+}$ (50 days old) during exposure to graded increases in $CO_2$. **A** Traces of raw and integrated ($\int$) diaphragm and abdominal EMG activity show that $Tcf4^{+/+}$ mice treated with saline (30 μL; I.P.) respond to 5 and 7% $CO_2$ with proportional increases in $Dia_{EMG}$ and $Abd_{EMG}$ activity. Conversely, saline (30 μL; I.P.) treated $Tcf4^{tr/+}$ mice show a diminished $Dia_{EMG}$ response to $CO_2$ and completely lacked $Abd_{EMG}$ activity, even at 7% $CO_2$. Systemic (I.P.) administration of PF-04531083 (40 mg/kg) increased $CO_2$-dependent $Dia_{EMG}$ but not $Abd_{EMG}$ activity in $Tcf4^{tr/+}$ mice. **B**, **C** Summary data show effects of $CO_2$ on $Dia_{EMG}$ amplitude (**B** $F_{2,10} = 40.74$, $p < 0.0001$, two-way ANOVA) and frequency (**C** $F_{2,10} = 27.96$, $p < 0.0001$, two-way ANOVA) in $Tcf4^{+/+}$ and $Tcf4^{tr/+}$ mice ($n = 6$ biologically independent animals/genotype, data are presented as mean values ± SEM). **D**, **E** summary data show effects of $CO_2$ on $Abd_{EMG}$ amplitude (**D** $F_{2,10} = 145.0$, $p < 0.0001$, two-way ANOVA) and frequency (**E** $F_{2,10} = 655.4$, $p < 0.0001$, two-way ANOVA) in $Tcf4^{+/+}$ and $Tcf4^{tr/+}$ mice ($n = 6$ biologically independent animals/genotype, data are presented as mean values ± SEM). These results are consistent with anatomical evidence that Phox2b+ neurons in the lateral parafacial region are severely depleted in $Tcf4^{tr/+}$ mice, and the possibility that these cells are a key determinant of expiratory activity (i.e., function as expiratory $pF_L$ neurons). Asterisk (*) indicate the different from 0% $CO_2$ within condition; #, different between genotypes/conditions. One symbol = $p < 0.05$, two symbols = $p < 0.01$, three symbols = $p < 0.001$, four symbols = $p < 0.0001$ (two-way RM-ANOVA with Tukey's multiple comparison test).

disrupted in $Tcf4^{tr/+}$ mice. For example, in control mice ($Phox2b^{Cre}::Ai14::Tcf4^{+/+}$ mice) Phox2b+ parafacial neurons that project to the pre-BötC primarily target Sst-positive neurons (97% of eYFP labeled puncta co-localized with Sst-IR) (Fig. 4E, F). Conversely, Phox2b parafacial neurons in $Phox2b^{Cre}::Ai14::Tcf4^{tr/+}$ animals project almost exclusively to Sst-negative pre-BötC neurons (96% of eYFP labeled puncta did not co-localize with Sst-IR) (Fig. 4E, F). Considering Sst+ pre-BötC neurons are the primary relay between inspiratory rhythmogenic elements of this region and motor output[36], alterations in this connectivity may contribute to respiratory disfunction in PTHS. Note, the distribution of Sst-positive pre-BötC neurons was similar between genotypes (419 vs. 439 Sst+ pre-BötC neurons in slices from $Tcf4^{+/+}$ and $Tcf4^{tr/+}$, respectively; $T_4 = 0.5858$, $p > 0.05$). These results suggest interactions between $Tcf4$ and $Atoh1$ contribute to cell-type specific deficits in PTHS.

$Tcf4$ is also expressed in postmitotic cells[37,38] where it regulates ion channel expression and neural excitability[6,26]; therefore, we wanted to determine whether chemosensitive RTN neurons still present in $Tcf4^{tr/+}$ mice function normally. To test this, we characterized the firing activity of chemosensitive RTN neurons in slices from neonatal $Tcf4^{+/+}$ and $Tcf4^{tr/+}$ mice under control conditions and during exposure to 10% $CO_2$. Chemosensitive RTN neurons were identified by their location, firing response to $CO_2$, and in some cases Phox2b immunoreactivity (Fig. 5B). Chemosensitive RTN neurons in slices from $Tcf4^{+/+}$ mice had an average baseline activity of $1.4 \pm 0.5$ Hz under control conditions (5% $CO_2$; pH 7.3) and increased their activity by $1.5 \pm 0.3$ Hz in response to 10% $CO_2$ ($pH_o = 7.0$) (Fig. 5A–D). This $CO_2/H^+$ response profile is similar to that which we[39] and others[40,41] have reported for chemosensitive RTN neurons in control tissue. While RTN neurons in slices from $Tcf4^{tr/+}$ mice were equally as active under control conditions (5% $CO_2$) ($1.7 \pm 0.2$ Hz, $T_{18} = 0.3118$, $p > 0.05$) (Fig. 5C), they showed a blunted firing response to 10% $CO_2$ ($0.7 \pm 0.1$ Hz, $T_{18} = 2.126$, $p = 0.0476$)

$Phox2b$-negative neurons remained similar in tissue from control and $Tcf4^{tr/+}$ mice, respectively (Supplementary Fig. 6). Together, these results suggest glutamatergic $Phox2b+$ parafacial neurons are selectively disrupted by loss of $Tcf4$. Also consistent with this, we found other Phox2b expressing populations in the caudal nucleus tractus solitarius (cNTS), locus coeruleus (LC) and facial motor nucleus, which are not dependent on $Atoh1$[33,34], showed normal Phox2b expression in tissue from $Tcf4^{tr/+}$ mice (Supplementary Fig. 5). Therefore, despite the widespread expression of $Tcf4$[35], loss of this transcription factor preferentially disrupts development of Phox2b+ neurons in the lateral parafacial region, and to the extent these cells function as expiratory $pF_L$ neurons; these results suggest $Tcf4$ regulates a $pF_L$-specific set of genes.

$Atoh1$ also regulates proper targeting of Phox2b+ parafacial neurons to the pre-BötC[22]; therefore, to characterize these connections in a PTHS model, we crossed $Tcf4^{+/+}$ and $Tcf4^{tr/+}$ mice with $Phox2b^{Cre}::Ai14$ animals and offspring of this cross (postnatal day 31) received bilateral parafacial injections of a Cre-dependent anterograde tracer (AAV2-Ef1α-DIO-hChR2(H134R)-EYFP; $4.2 \times 10^{12}$ molecules/mL). Consistent with a net loss of RTN neurons (Fig. 4C, D), we found the average number of labeled pre-BötC neurons decreased from 144 cells per control mouse to 112 cells per $Tcf4^{tr/+}$ mouse ($T_4 = 5.168$, $p = 0.0067$; Fig. 4E, F). We also found that cell type specific targeting of RTN projections to the pre-BötC was

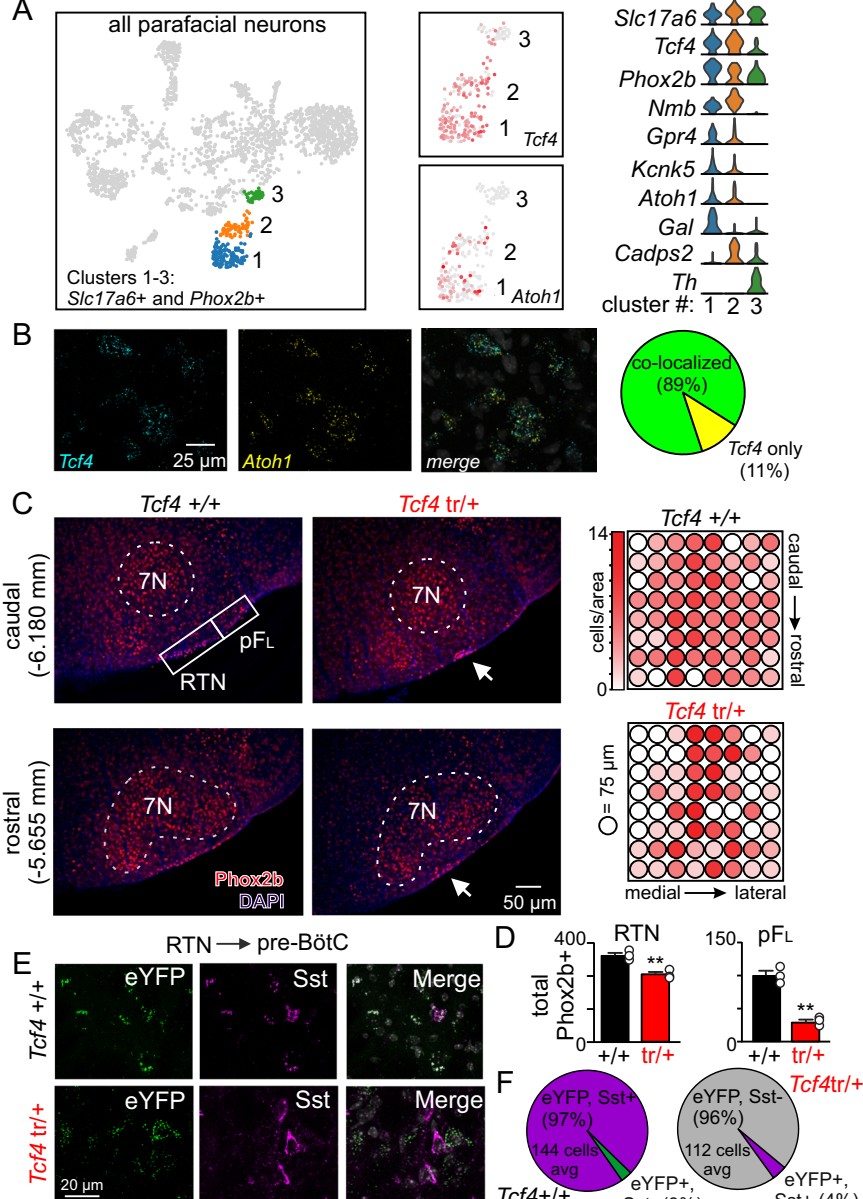

**Fig. 4 Morphological and projection abnormalities of Phox2b+ parafacial neurons in $Tcf4^{tr/+}$ mice. A** Left side, t-distributed stochastic neighbor embedding (t-SNE) plot shows the single-cell transcriptome for ventral parafacial neurons; cell types that co-express both *Slc17a6* and *Phox2b* are color coded by cluster (cluster 1 is blue, cluster 2 is orange, and cluster 3 is green). Middle, UMAP plots showing expression of *Tcf4* (top) and *Atoh1* (bottom) in sub-clusters of *Slc17a6+* and *Phox2b+* neurons. Right, violin plots show cluster-specific differential gene expression (gene expression from 0 to 4 counts/ cell is on the *y*-axis). Clusters 1–2 are putative RTN chemoreceptors based on expression of *Phox2b, Nmb, Gpr4,* and *Kcnk5*. Cluster 3 shows a profile consistent with C1 pre-sympathetic neurons including tyrosine hydroxylase (*Th*) and *Phox2b* but not *Gpr4, Kcnk5,* or *Nmb*[31]. *Tcf4* is expressed by clusters 1–3 but only co-localized with *Atoh1* in clusters 1–2. **B** Coronal sections from a $Tcf4^{+/+}$ mouse show parafacial neurons that express *Tcf4* transcripts (cyan) and *Atoh1* transcripts (yellow). Right, summary of fluorescent in situ hybridization results ($n = 3$, 12 days of postnatal) show that 89% of *Tcf4* labeling in the parafacial region co-localized with *Atoh1* labeling (green indicates co-labeled *Tcf4* and *Atoh1* cells, yellow indicates *Tcf4* transcript only). **C** Photomicrographs of coronal sections from a $Tcf4^{+/+}$ and $Tcf4^{tr/+}$ mouse show Phox2b-immunoreactivity (Phox2b-IR, red) in the caudal (top) and rostral (bottom) parafacial regions (values denote distance behind bregma, co-localized with blue DAPI signal). Regions of interest penetrated ~75 μm dorsally from the ventral surface and spanned 600 μm medially from the trigeminal, the lateral most 150 μm was considered the pF_L. Right: Summary data ($n = 3$ mice/genotype) show the distribution of Phox2b-IR soma across the caudal to rostral (*y* axis; eight slices total per animal) and medial to lateral (*x*-axis) extent of the parafacial region. **D** Summary data show that Phox2b-IR is diminished in the pF_L ($T_4 = 8.510$, $p = 0.0010$) and to a lesser extent in the RTN ($T_4 = 5.439$, $p = 0.0055$) from $Tcf4^{tr/+}$ mice ($n = 5$ biologically independent animals/genotype, data are presented as mean values ± SEM. Also note that Phox2b-IR neurons tended to clump in the medial parafacial region of $Tcf4^{tr/+}$ (arrow). **E**, AAV2-Ef1α-DIO-hChR2(H134R)-EYFP was injected bilaterally into the medial parafacial region of $Phox2b^{Cre}$::Ai14::$Tcf4^{+/+}$ (control) and $Phox2b^{Cre}$::Ai14::$Tcf4^{tr/+}$ mice and labeled puncta were imaged in the pre-BötC. Photomicrographs and summary data (**F**, left) ($n = 3$ mice/genotype) show in control tissue that most (97%) green-labeled puncta make close associations with Sst-IR (purple in images and pie chart) pre-BötC neurons. Conversely, tissue from $Phox2b^{Cre}$::Ai14::$Tcf4^{tr/+}$ mice shows the opposite labeling pattern; 96% of green-labeled puncta (bottom) do not co-localize with Sst-labeled pre-BötC neurons (**F** right, gray area in pie chart). **$p < 0.01$ (unpaired *t*-test).

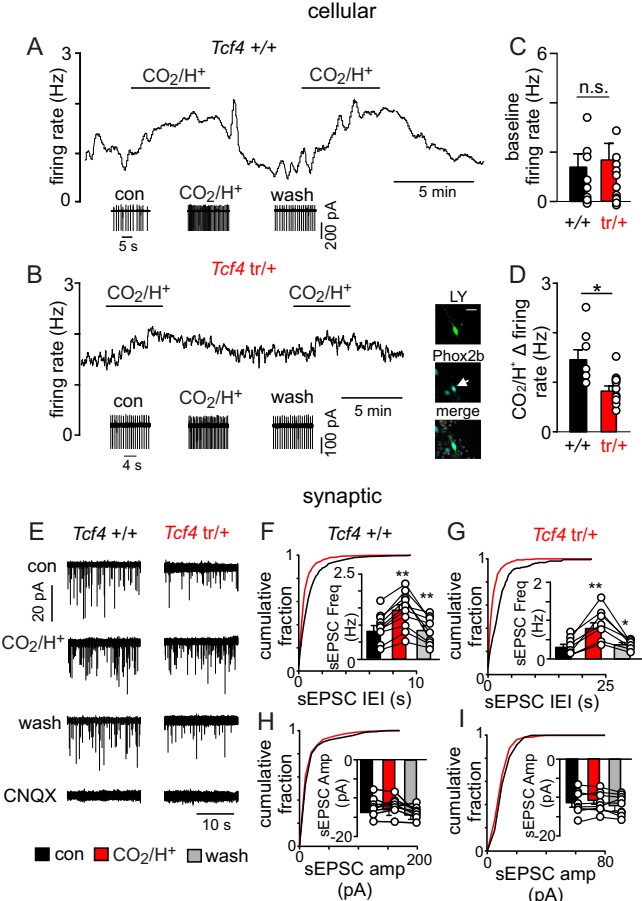

**Fig. 5 Chemosensitive RTN neurons in slices from $Tcf4^{tr/+}$ mice show reduced $CO_2/H^+$ sensitivity at the cellular and synaptic levels. A, B** Traces of firing rate and segments of holding current from chemosensitive RTN neurons in slices from control (**A**) and $Tcf4^{tr/+}$ (**B**) mice show typical levels of activity for each genotype under control conditions (5% $CO_2$, pH 7.3) and during exposure to 10% $CO_2$ (pH 7.0). Inset, double-immunolabeling shows that a Lucifer Yellow-filled $CO_2/H^+$ sensitive neuron (green) in a slice from a $Tcf4^{tr/+}$ mouse is immunoreactive for Phox2b (cyan). We confirmed that 6/6 $CO_2/H^+$ activated neurons in slices from $Tcf4^{tr/+}$ mice are Phox2b-IR. Scale bar = 25 μm. **C, D** Summary data ($n = 13$ cells/genotype) shows that RTN chemoreceptors in slices from $Tcf4^{tr/+}$ exhibit normal activity under baseline conditions (**C** $n = 13$ cells/genotype, eight biologically independent animals/genotype, $T_{24} = 0.3118$, $p > 0.05$, data are presented as mean values ± SEM) but have a reduced firing response to 10% $CO_2$ (**D** $n = 9$ cells/genotype, five biologically independent animals/genotype, $T_{17} = 2.378$, $p = 0.0294$, data are presented as mean values ± SEM). **E** Traces of holding current ($I_{hold} = -60$ mV) from chemosensitive RTN neurons in slices from $Tcf4^{+/+}$ and $Tcf4^{tr/+}$ mice shows spontaneous excitatory synaptic current (sEPSC) events under control conditions and during exposure to 10% $CO_2$ or CNQX (10 μM). **F, G** summary ($n = 8$ cells/genotype, five biologically independent animals/genotype, data presented as mean values ± SEM) cumulative distribution plots of sEPSC inter-event interval (bin size: 250 ms) and bar graphs of mean sEPSC frequency under each experimental condition show that sEPSC frequency was diminished in $Tcf4^{tr/+}$ under control conditions ($T_{15} = 2.417$, $p = 0.0289$) and during exposure to 10% $CO_2$ ($T_{15} = 3.126$, $p = 0.0061$) compared to control. **H, I** Cumulative distribution plots of sEPSC amplitude (bin size: 5 pA) and bar graphs of mean sEPSC amplitude under each condition show that $CO_2$ minimally affects sEPSC amplitude in either genotype (control: $n = 10$ biologically independent animals, $F_{2,25} = 0.042$, $p > 0.05$, $Tcf4^{tr/+}$: $n = 9$ biologically independent animals, $F_{2,25} = 0.386$, $p > 0.05$, data are presented as mean values ± SEM). *$p < 0.05$, **$p < 0.01$ (paired t-test for comparison in **D** and paired one-way ANOVA for **F–I**).

(Fig. 5D). These results show that $Tcf4$ haploinsufficiency diminished cellular excitability of RTN chemoreceptors.

We also characterized baseline and $CO_2/H^+$-dependent excitatory and inhibitory synaptic inputs to chemosensitive RTN neurons. Once a $CO_2/H^+$ activated neuron was identified, we obtained whole-cell access and in voltage-clamp, recorded spontaneous excitatory or inhibitory postsynaptic currents (sEPSCs and sIPSCs) in relative isolation by holding voltage near the reversal potential for IPSCs ($-60$ mV) or EPSCs (0 mV), respectively (Fig. 5E–I). Consistent with evidence that RTN neurons talk to each other through $CO_2/H^+$-dependent excitatory interactions[42], and our anatomical results suggesting loss of $Tcf4$ results in fewer chemosensitive RTN neurons (Fig. 4B), we found that RTN neurons in slices from $Tcf4^{tr/+}$ mice showed a lower sEPSC frequency under baseline conditions (0.62 Hz $Tcf4^{+/+}$ vs. 0.40 Hz $Tcf4^{tr/+}$) ($T_{15} = 2.417$, $p = 0.0289$) and in response to 10% $CO_2$ (0.84 Hz $Tcf4^{+/+}$ vs. 0.33 Hz $Tcf4^{tr/+}$) ($T_{15} = 3.126$, $p = 0.0061$) compared to RTN neurons in slices from $Tcf4^{+/+}$ animals (Fig. 5E–G). Amplitude of sEPSCs was similar between genotypes and experimental conditions ($T_{13} = 0.3535$, $p > 0.05$; Fig. 5H–I). Consistent with evidence that inhibitory synaptic inputs contribute to RTN chemoreception[42], we found that exposure to 10% $CO_2$ decreased sIPSC frequency and amplitude (Supplementary Fig. 7); however, it did so by similar amounts in both genotypes ($T_9 = 0.6810$, $p > 0.05$). Together, these results show that chemosensitive RTN neurons are disrupted at the cellular and network level in $Tcf4^{tr/+}$ mice, and therefore likely contribute to the diminished chemoreflex observed in $Tcf4^{tr/+}$ mice.

**Nav1.8 channels are therapeutic targets in PTHS.** Recent evidence showed Nav1.8, a sodium channel normally restricted to sensory nerves[43], was ectopically expressed centrally in $Tcf4^{tr/+}$ mice where it disrupted repetitive firing behavior of cortical neurons by a mechanism involving depolarizing block[6]. It was also shown that bath application of a selective Nav1.8 channel blocker rescued this firing deficit[6], thus identifying Nav1.8 as a target for treatment of PTHS-associated behavioral deficits. To explore this possibility at the level of the RTN, we first wanted to determine whether $Scn10a$ transcript is upregulated in residual Phox2b+ parafacial neurons in $Tcf4^{tr/+}$ mice. For this experiment, we isolated the ventral parafacial region from $Phox2b^{Cre}::Ai14::Tcf4^{+/+}$ and $Phox2b^{Cre}::Ai14::Tcf4^{tr/+}$ mice and performed fluorescence activated cell sorting to obtain an enriched populations of TdT-positive cells from each genotype for subsequent targeted qPCR analysis of $Scn10a$ transcript. Consistent with evidence from other brain regions[6,38], we found detectable levels of $Scn10a$ transcript in Phox2b+ neurons from adult $Tcf4^{tr/+}$ mice (average raw Ct value: 30.1 ± 0.2; ΔCt value: 11.8 ± 0.1; $n = 3$ animals/genotype, three technical replicates per sample; Taqman probe: Mm01342502_g1) but not age matched $Tcf4^{+/+}$ control animals (transcript not detected; $Gapdh$ average raw Ct value: 17.2 ± 0.3; $n = 3$ animals/genotype; three technical replicates per sample).

To determine whether aberrant expression of $Scn10a$ in Phox2b neurons disrupts cellular function, we used slice-patch electrophysiology to characterized baseline activity and $CO_2/H^+$-sensitivity of chemosensitive RTN neurons in $Tcf4^{tr/+}$ under control conditions and in the presence of a Nav1.8 blocker (PF-04531083; 1 μM). We found that bath application of PF-04531083 stimulated baseline activity by ~50% ($T_6 = 5.729$, $p = 0.0012$; Fig. 6A, B) and increased the average firing response

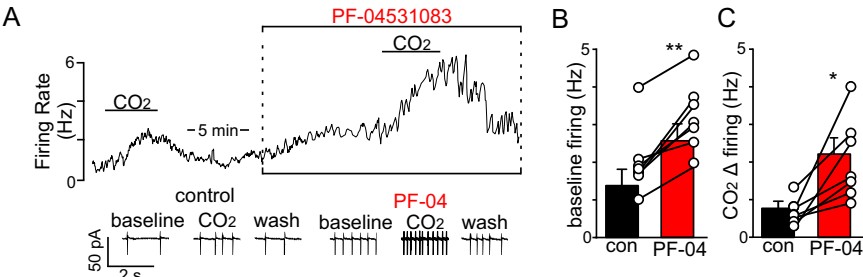

**Fig. 6 Pharmacological blockade of Nav1.8 increases baseline activity and $CO_2/H^+$-sensitivity of RTN chemoreceptors in slices from $Tcf4^{tr/+}$.** **A** Trace of firing rate and segments of holding current from chemosensitive RTN neuron in a slice from a $Tcf4^{tr/+}$ mouse shows that bath application of PF-04531083 (abbreviated PF-04; 1 μM) increased baseline activity and the firing response to 10% $CO_2$. **B, C** Summary data ($n = 7$ cells) shows that PF-04531083 increased baseline activity by ~1 Hz (**B**) ($T_6 = 5.729$, $p = 0.0012$, data are presented as mean values ± SEM) and nearly doubled the firing response to $CO_2$ (**C**) ($T_6 = 3.669$, $p = 0.0105$, data are presented as mean values ± SEM). Asterisk (*) indicate different from control at $p < 0.05$ (one symbol) or $p < 0.01$ (two symbols) (paired $t$-test).

to 10% $CO_2$ from 0.8 ± 0.1 to 2.2 ± 0.4 Hz ($T_6 = 3.669$, $p = 0.0105$; Fig. 6A–C). These results suggest residual Phox2b+ neurons located predominantly in the medial parafacial region (Fig. 4B) (putative RTN chemoreceptors) can be targeted to improve respiratory function in $Tcf4^{tr/+}$.

To test this possibility at the behavioral level, we measured baseline respiratory activity in adult $Tcf4^{tr/+}$ mice before and after systemic (I.P.) administered of the blood brain barrier permeable Nav1.8 blocker PF-04531083 (40 mg/kg; solubility 0.7 μg/mL, hNav1.8 EC$_{50}$ 190 nM[44]). We found that PF-04531083 treatment improved baseline breathing by decreasing episodes of hyperventilation ($T_5 = 5.168$, $p = 0.0036$; Fig. 7A, B), increasing the occurrence of spontaneous sighs ($T_5 = 2.825$, $p = 0.0369$; Fig. 7D, E) and diminishing duration of post-sigh apnea ($T_5 = 4.885$, $p = 0.0045$; Fig. 7D, F). Conversely, administration of PF-04531083 minimally effected baseline breathing in $Tcf4^{+/+}$ mice (Fig. 7A–C). We also found that PF-04531083 improved the $CO_2$ minute ventilatory response of $Tcf4^{tr/+}$ mice (Fig. 8A, B) (0–7% $CO_2$ slope: 0.59 ± 0.08 saline vs. 0.79 ± 0.10 PF-04531083; $p = 0.0314$) to an amount similar to $Tcf4^{+/+}$ (0–7% $CO_2$ slope: 1.11 ± 0.1; $p > 0.05$). Interestingly, systemic (I.P.) application of a blood brain impermeable Nav1.8 blocker (PF-06305591; 2 mg/kg; solubility 2 mg/ml, hNav1.8 EC$_{50}$ 15 nM[44]) did not improve the ventilatory response to $CO_2$ in $Tcf4^{tr/+}$ mice (0–7% $CO_2$ slope: 0.56 ± 0.06 saline vs. 0.50 ± 0.09 PF-06305591; $p > 0.05$) (Fig. 8C). Furthermore, since breathing problems in PTHS typically manifest during childhood[3], we also characterized the chemoreflex in neonatal pups (11–12 days old) under control conditions and after administration of Nav1.8 blockers. We found that $Tcf4^{tr/+}$ pups receiving PF-04531083 (40 mg/kg) showed a robust increase in respiratory activity (62.5% increase in minute ventilation from 3 to 7% $CO_2$) to an amount that was similar to control mice that received saline or PF-04531083 ($F_{1,4} = 6.540$, $p = 0.0628$; Supplementary Fig. 8).

To determine whether Nav1.8 channels expressed by residual Phox2b+ parafacial neurons can be targeted to improve breathing in $Tcf4^{tr/+}$, we used a viral delivery system to express a short hairpin RNA against $Scn10a$ (gene encoding Nav1.8) in Cre-recombinase-dependent manner in Phox2b+ parafacial neurons in $Tcf4^{tr/+}$ mice. Specifically, we injected AAV2-PRSx8-eGFP-mScn10a-shRNAmir (10 nL/side, Vector Biolabs) bilaterally into the medial parafacial region of $Phox2b^{Cre}$::Ai14::$Tcf4^{tr/+}$ mice ($Tcf4^{tr/+}$ $Scn10a$ shRNA mice) (Supplementary Fig. 9). Two weeks after virus injections, we confirmed that 81% of TdT-labeled Phox2b+ parafacial neurons are GFP+ (Supplementary Fig. 9A). However, a modest amount of viral-mediated GFP labeling (11%) did not colocalize with reporter expression (Supplementary Fig. 9A). We assessed respiratory activity before

and two weeks after viral injections and found that Phox2b-specific knockdown of $Scn10a$ improved $CO_2/H^+$-dependent respiratory output (0–7% $CO_2$ slope: 0.33 ± 0.04 before injection vs. 0.56 ± 0.07 after injection; $p = 0.0491$; Supplementary Fig. 9C). These results suggest that Nav1.8 in residual Phox2b+ parafacial neurons contributes to breathing problems in PTHS. By similar logic, if Phox2b+ neurons in the lateral parafacial region mediate active expiration during high $CO_2$, and since these neurons are largely absent in $Tcf4^{tr/+}$ mice (Fig. 2B), then PF-04531083 is not expected to rescue $CO_2/H^+$-dependent expiratory activity in $Tcf4^{tr/+}$ mice. This is exactly what we found; under isoflurane anesthesia adult $Tcf4^{tr/+}$ mice that received PF-04531083 showed minimal abdominal electromyography (EMG) activity even at 7% $CO_2$ (Fig. 8D). Consistent with observations in awake mice, we found that PF-04531083 improved inspiratory activity in anesthetized $Tcf4^{tr/+}$ mice by increasing diaphragm EMG amplitude at 5 and 7% $CO_2$ ($F_{1,5} = 119.6$, $p < 0.0001$) and frequency at 7% $CO_2$ ($T_{10} = 2.439$, $p = 0.0349$; Fig. 8E, F).

We also found that other behavioral abnormalities associated with PTHS were also improved by systemic blockade of Nav1.8 channels. For example, PF-04531083 ($T_7 = 4.866$, $p = 0.0018$) but not PF-06305591 ($T_7 = 0.5667$, $p > 0.05$) reduced hyperactivity of $Tcf4^{tr/+}$ mice to levels similar to littermate control mice (Fig. 9A, B). Therefore, the therapeutic utility of targeting Nav1.8 is not limited to the respiratory system but rather may improve multiple features of PTHS, suggesting that it is a therapeutic target with broad clinical utility.

## Discussion

Breathing problems are a common but poorly understood feature of PTHS. Here, we show that a $Tcf4$ truncation mouse model of PTHS recapitulates the prevalence and periodic nature of the apneic phenotype observed in PTHS patients under room air conditions. We also show that $Tcf4^{tr/+}$ mice have a compromised ability to regulate breathing in response to $CO_2$ (Fig. 2G, H), suggesting altered chemoreceptor function contributes to breathing problems in PTHS. Consistent with this and previous evidence that $Tcf4$ interacts with $Atoh1$ to regulate development of brainstem respiratory centers[24], we found that Phox2b+ parafacial neurons including RTN chemoreceptors were depleted in $Tcf4^{tr/+}$ mice (Fig. 4C), and remaining RTN chemoreceptors in $Tcf4^{tr/+}$ mice showed diminished $CO_2/H^+$-responsiveness (Fig. 5) and altered connectivity with the pre-BötC (Fig. 4E). Note also that $Nmb$-expressing parafacial neurons, which includes Phox2b+ RTN neurons[31], regulate sighing[45], thus loss of this population in $Tcf4^{tr/+}$ mice is consistent with diminished sigh activity. Further, we show that Nav1.8 channels are

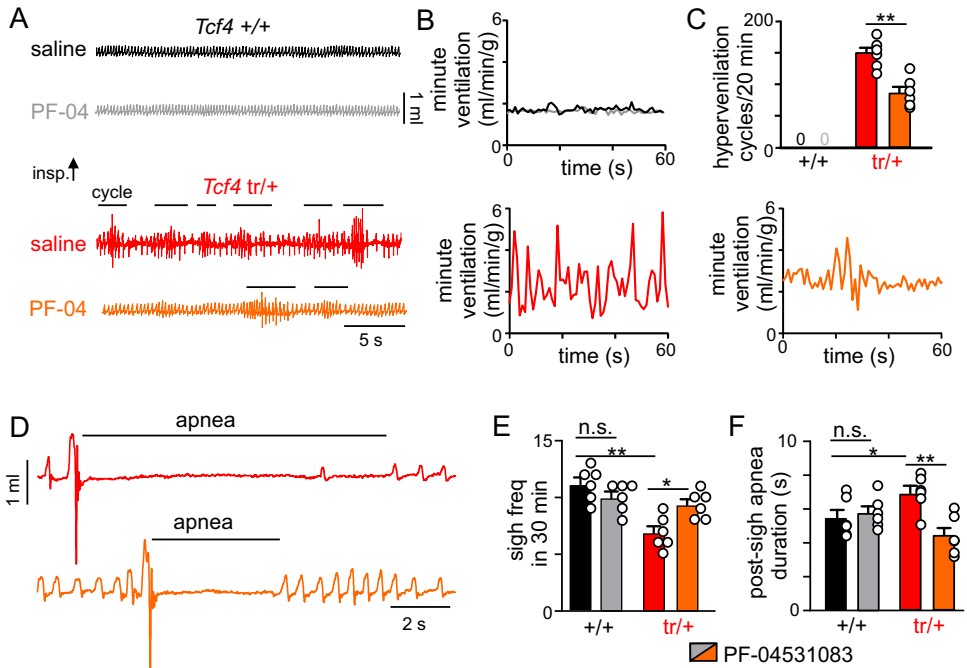

**Fig. 7 Systemic application of a Nav1.8 blocker improved baseline breathing in _Tcf4_$^{tr/+}$ mice.** For these experiments we characterized respiratory activity in _Tcf4_$^{+/+}$ and _Tcf4_$^{tr/+}$ mice ~1.5 h after systemic (I.P.) administration of saline (30 μL) or PF-04531083 (abbreviated PF-04; 40 mg/kg, a selective Nav1.8 channel blocker that crosses blood brain barrier). **A** Traces of respiratory activity under room air conditions show that _Tcf4_$^{tr/+}$ mice that received PF-04531083 (orange) exhibit fewer cycles of hyperventilation (designated by a horizontal line) compared to those that received saline (red). Conversely, control mice showed stable respiratory activity following injections of saline (black) or PF-04531083 (gray). **B** Traces of minute ventilation (same animals as **A**) show the pattern of respiratory activity after saline or PF-04531083 injections. Note that PF-04531083 stabilized breathing in _Tcf4_$^{tr/+}$ compared to saline. **C** Summary plot shows that PF-04531083 treatment decreased the number of hyperventilation cycles exhibited by _Tcf4_$^{tr/+}$ mice ($n = 6$ biologically independent animals/genotype, $T_5 = 5.168$, $p = 0.0036$, data are presented as mean values ± SEM). **D–F** Respiratory traces (**D**) and summary data show that PF-04531083 increased sigh frequency (**E** $n = 6$ biologically independent animals/genotype, $T_5 = 2.825$, $p = 0.0369$, data are presented as mean values ± SEM) and reduced the duration of post-sigh apnea (**F** $n = 6$ biologically independent animals/genotype, $T_5 = 4.885$, $p = 0.0045$, data are presented as mean values ± SEM) in _Tcf4_$^{tr/+}$ under room air conditions. Asterisk (*) indicate the difference between genotypes (unpaired $t$-test) at $p < 0.05$ (one symbol) or $p < 0.01$ (two symbols).

ectopically expressed by RTN chemoreceptors in _Tcf4_$^{tr/+}$ mice, and systemic application of a Nav1.8 channel blocker improved chemoreceptor function at the cellular (Fig. 6) and behavioral (Figs. 7–9) levels in _Tcf4_$^{tr/+}$ mice. This result is surprising considering increased expression of Nav1.8 is expected to promote rather than limit neural excitability. Although mechanisms underlying this response are not clear, since Nav1.8 channels in peripheral neurons are slowly inactivating[46], we speculate that expression of these channels in Phox2b neurons will limit excitability in part by decreasing input resistance and shunting synaptic or $CO_2/H^+$-induced potentials. However, this possibility requires further investigation. Together, these pre-clinical results identify parafacial Phox2b+ neurons as a contributing factor to disordered breathing in PTHS, and establish Nav1.8 channels as a high priority target with therapeutic potential in this disease.

The lateral parafacial region is thought to regulate active expiration[19,47]; however, previous work suggests pF$_L$ neurons do not express Phox2b[30] and unique genetic markers delineating this population are lacking. Here, we show that _Tcf4_$^{tr/+}$ mice fail to generate active expiration during exposure to 7% $CO_2$ (Fig. 3), even after systemic application of a Nav1.8 blocker that improved other aspects of breathing including RTN chemoreception (Fig. 8). We also show that the proportion of Phox2b+ neurons in the lateral parafacial region was selectively reduced in _Tcf4_$^{tr/+}$ mice compared to the broader population of glutamatergic parafacial neurons (Supplementary Fig. 6) or other Phox2b+ populations (Fig. 4C). These results suggest expiratory

pF$_L$ neurons express Phox2b. However, this possibility requires further investigation and so at present it remains unclear whether expiratory pF$_L$ neurons are an extension of the RTN or comprise a functionally discrete respiratory center.

Breathing problems in PTHS are similar a related disorder called Rett syndrome (RTT), an autism spectrum disorder caused by mutations in the methyl-CpG-binding protein 2 (MeCP2)[48]. For example, both syndromes share several differentially expressed genes[38] and like in PTHS patients[29], breathing problems in RTT are associated with unstable periodic breathing characterized by alternating bouts of hyperventilation followed by hypoventilation[49], and perhaps most strikingly, breathing problems in PTHS and RTT share a similar wake-dependence[3,29,49]. This is unusual because most respiratory problems including those associated diminished RTN chemoreception typically manifest during sleep[7]. These results suggest breathing problems in PTHS and RTT may involve multiple cell types and several brain regions. However, similar to our findings in _Tcf4_$^{tr/+}$ mice, evidence from mouse models of RTT suggest disordered breathing in this condition results in part from diminished RTN chemoreception[11,12]. Interestingly, astrocytes appear to contribute to chemoreceptor dysfunction in RTT. For example, astrocyte-specific MeCP2-deficient mice fail to elicit a normal ventilatory response to $CO2$[11] and RTN astrocytes show a diminished $Ca^{2+}$ responses to acidification[50]. Although we did not detect _Tcf4_ in parafacial astrocytes from control animals and astrocytes in _Tcf4_$^{tr/+}$ mice appeared morphologically normal; nevertheless, it remains

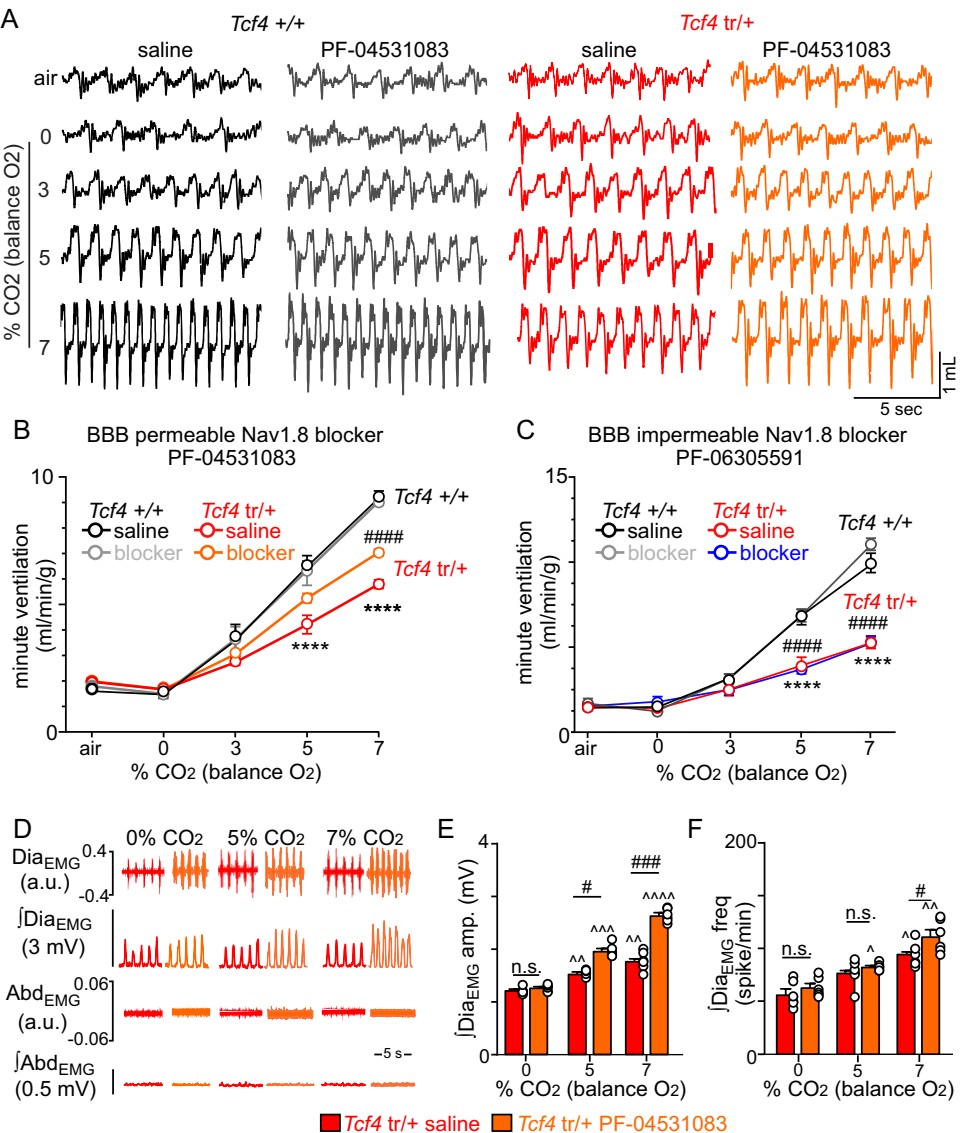

**Fig. 8 Central Nav1.8 channels can be targeted to improve $CO_2/H^+$-dependent respiratory activity in $Tcf4^{tr/+}$ mice. A** Traces of respiratory activity from saline (black) or PF-04531083 (gray) treated $Tcf4^{+/+}$ and saline (red) or PF-04531083 (orange) treated $Tcf4^{tr/+}$ mice during exposure to room air, 100% $O_2$ and 3–7% $CO_2$ (balance $O_2$). **B** Summary plots of minute ventilation show that PF-04531083 improved $CO_2$-dependent respiratory output in $Tcf4^{tr/+}$ mice (0–7% $CO_2$ slope: 0.59 ± 0.08 saline vs. 0.79 ± 0.10 PF-04531083; $n = 5$ biologically independent animals, $p = 0.0314$, data are presented as mean values ± SEM) to a level not different from $Tcf4^{+/+}$ mice (0–7% $CO_2$ slope: 1.11 ± 0.1; $n = 5$ biologically independent animals, $p > 0.05$, data are presented as mean values ± SEM). **C** Summary plots of minute ventilation show that systemic application of PF-06305591 (2 mg/kg, a selective Nav1.8 channel blocker that does not readily cross the blood brain barrier) minimally effected respiratory activity in both genotypes (0–7% $CO_2$ slope: 0.56 ± 0.06 saline vs. 0.50 ± 0.09 PF-06305591; $n = 5$ biologically independent animals/genotype, $p > 0.05$, data are presented as mean values ± SEM) ($Tcf4^{tr/+}$ mice injected with Pf-06305591 are indicated with blue). **D** Traces of raw and integrated ($\int$) diaphragm and abdominal EMG activity show that $Tcf4^{tr/+}$ mice treated with saline (30 μL; I.P.) show a diminished $Dia_{EMG}$ response to $CO_2$ and completely lacked $Abd_{EMG}$ activity, even at 7% $CO_2$. Systemic (I.P.) administration of PF-04531083 (40 mg/kg) increased $CO_2$-dependent $Dia_{EMG}$ but not $Abd_{EMG}$ activity in $Tcf4^{tr/+}$ mice. **E, F** Summary data show effects of $CO_2$ on $Dia_{EMG}$ amplitude (**E** $n = 6$ biologically independent animals/genotype, data are presented as mean values ± SEM) and frequency (**F** $n = 6$ biologically independent animals/genotype, data are presented as mean values ± SEM) in $Tcf4^{tr/+}$ mice that received saline or PF-04531083. These results are entirely consistent with the respiratory phenotype exhibited by awake $Tcf4^{tr/+}$ mice under control conditions and after PF-04531083 treatment (Fig. 7). Asterisk (*) indicate the different between genotypes; #, used to distinguish difference between PF-04531083 injected mice of both genotypes. ^, different from 0% $CO_2$ within condition ($Dia_{EMG}$). One symbol = $p < 0.05$, two symbols = $p < 0.01$, three symbols = $p < 0.001$, four symbols = $p < 0.0001$ (two-way RM-ANOVA with Tukey's multiple comparison test).

possible that RTN astrocytes contribute to chemoreceptor dysregulation in PTHS. This possibility warrants further investigation.

In sum, our results identify $Tcf4$ as a requisite determinant of development and function of parafacial respiratory centers. We also provide novel mechanistic insight into causes and treatment of disordered breathing in PTHS by targeting Nav1.8 channels.

## Methods

**Animals.** All procedures were performed in accordance with National Institutes of Health and University of Connecticut Animal Care and Use Guidelines. All animals were housed in a 12:12 light/dark cycle (average ambient temperature 72 °F, average humidity 50%) with unlimited access to normal chow and an enrichment hutch. No other items were placed in home cages. $Phox2b^{Cre}$::TdT (Ai14) (JAX # 016223 and 007914) were crossed to $Tcf4^{tr/+}$ mice to quantify aberrant $Scn10a$ expression in the RTN and for anterograde tracing experiments. $Tcf4^{tr/+}$ mice were also crossed with

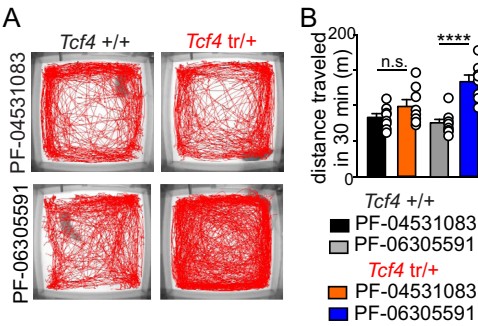

**Fig. 9 Central Nav1.8 channels can be targeted to improve locomotor abnormalities in *Tcf4*$^{tr/+}$ mice. A** Locomotor activity maps from *Tcf4*$^{+/+}$ and *Tcf4*$^{tr/+}$ mice treated with saline, PF-04531083, or PF-06305591, movement was recorded for 30 min following placement in a novel open field arena. **B** Summary plots for distance traveled over 30 min depicts *Tcf4*$^{+/+}$ mice injected with PF-04531083 (black) or PF-06305591 (gray) and *Tcf4*$^{tr/+}$ mice injected with PF-04531083 (orange) or PF-06305591 (blue). These data show that *Tcf4*$^{tr/+}$ treated with PF-04531083 exhibited locomotor activity similar to *Tcf4*$^{+/+}$ ($n = 8$ biologically independent animals/genotype, $T_{14} = 1.452$, $p > 0.05$, data are presented as mean values ± SEM), whereas those that received PF-06305591 remained hyperactive compared to either experimental group ($n = 8$ biologically independent animals/genotype, $T_{14} = 5.492$, $p < 0.0001$, data are presented as mean values ± SEM). Note that PF-04531083 minimally affected respiratory or locomotor activity in control mice. Asterisk (*) indicate the different between injected mice of the same genotype (unpaired *t*-test). One symbol = $p < 0.05$, two symbols = $p < 0.01$, three symbols = $p < 0.001$, four symbols = $p < 0.0001$.

*Gfap*$^{Cre/ERT2}$ (JAX # 012849) to assess parafacial astrocyte properties in this model. The *Tcf4*$^{tr/+}$ line (JAX # 013598)[25] used in this study was inbred from the F1 generation at time of cryorecovery and maintained on a 50% 129Sl/J, 50% C57BL6/J mixed background. There were no alterations in background strain throughout breeding or between different breeding schemes (see Supplementary Fig. 1). Pups (mice ~P21 and below) were housed with both parents before experimentation and, in the case of in vivo experiments, during drug incubation periods between experiments. No pup was out of the home cage for more than 1 h during in vivo trials.

**Animal model characterization.** Body weights were measured using a tabletop scale to the tenth gram for newborn animals up to 60 days of age. If an animal was found dead or struggling to survive, weight was taken post-mortem prior to genotyping. For generation of a survival curve, mice were bred according to Breeding Strategy #2 (Supplementary Fig. 1). Pups that were stillborn or died shortly after birth were immediately weighed and genotyped. In this way, the *Tcf4*$^{tr/tr}$ survival curve accounts for immediate postpartum death/stillbirth or failure to initiate breathing with a 70% survival point at P0.

*Primers for genotyping.* *Tcf4* mice were genotyped per Jackson Laboratories website descriptions. In brief, two reactions were used to identify *Tcf4*$^{+/+}$, *Tcf4*$^{tr/+}$, and *Tcf4*$^{tr/tr}$ genotypes (Supplementary Fig. 1). Reaction A used common forward primer and splice variant reverse primer to determine if the truncation allele was present, while reaction B used the common forward primer and normal reverse primer to determine if wild type allele was present (primer sequence detailed in Supplementary Table 1).

*Comprehensive lab animal monitoring (CLAMS).* Metabolic monitoring O$_2$ consumption (VO$_2$) and CO$_2$ production (VCO$_2$) was performed using comprehensive lab animal monitoring systems (CLAMS, Columbus Instruments). Adult mice were individually housed on a 12:12 light:dark cycle in plastic cages with a running wheel, regular bedding, and regular chow for one week before experimentation. Three days before the metabolic experiment, each animal was placed in the CLAMS housing cage with metered water and waste collection. Mice had two days to acclimate to the metabolic chamber; on the third day, all results were recorded for a continuous 24 h period (Oxymax v5.54). After data collection, all raw results were exported and averaged out per hour, only including times of no wheel activity as assessed by an activity monitoring system within the CLAMS housing cage. Then, light and dark periods were determined and averaged per animal for statistical analysis. Both sexes were equally represented in the data set.

*Blood gas analysis.* Arterial blood gasses were collected from adult mice 6 weeks of age and older (>30 g) as previously reported[51]. In short, a RAPIDLab® 348 blood

gas analyzer (Siemens) was used for all blood gas analysis; all calibrations, QC, and use were performed as indicated by the manufacturer. Animals were anesthetized with an induction dose of 3% isoflurane and then quickly switched to 1% isoflurane for the remainder of arterial blood collection. The left carotid artery was exposed and quickly cannulated to allow for arterial blood to be collected and analyzed by the blood gas analyzer; no more than 5 s was spent between blood collection and analysis on the blood gas analyzer.

**Drugs.** PF-04531083 was suspended in corn oil and delivered in vivo at 40 mg/kg, as previously described[42]. For in vitro experiments, PF-04531083 was dissolved in DMSO and used at a concentration of 1 μM. PF-06305591 was dissolved in corn oil and used for in vivo experiments at 2 mg/kg, as previously described[38]. All drug concentrations were chosen based on the minimum dose required to affect primary behavior measures (e.g., response to painful stimuli)[44,52]. Note that PF-06305591 is more soluble and has a 10-fold lower EC50 compared to PF-04531083[44].

**Fluorescent in situ hybridization.** Twelve day old pups of each genotype were anesthetized with isoflurane, decapitated, and brainstem tissues were rapidly frozen with dry ice and 75% ethanol. Brainstem slices (14 μm thick) containing the RTN were cryosectioned, collected onto SuperFrost Plus microscope slides, and dried for at least 24 h at −20 °C. Slices were then fixed with 4% paraformaldehyde and dehydrated with 50, 70, and 100% ethanol. Fluorescent in situ hybridization was performed according to the RNAscope Multiplex Fluorescent Assay (ACD Cat# 320850) instructions; the probes used in our study were designed and validated by ACD and include: Mm-Atoh1 (Cat# 408791), Mm-Tcf4 (Cat# 423691), Mm-Phox2b (Cat# 407861), Mm-Slc17a6 (Cat# 319171), and Mm-Slc32a1 (Cat# 319191). Confocal images of FISH experiments were obtained using a Leica TSC SP8 at 1024 × 1024 resolution, with a digital zoom of 1.68, and a minimum Z stack of 10 μm. Confocal image files containing image stacks were loaded into ImageJ v2.0.0 for analysis.

*Cell counting analysis.* Images were loaded into the ImageJ software and all Z-stacks were merged to maximum intensity and all channels were split from the merged image. Cells were counted on each channel individually using the DAPI channel to distinguish an individual cell from background. Any cell that was partially out of frame was not included in the analysis. We used a threshold of five individual puncta to be considered positive.

**Immunohistochemistry.** Weaned mice (P21 and above) were anesthetized with 3% isoflurane and transcardially perfused with 20 mL of room temperature phosphate buffered saline (PBS, pH 7.4) followed by chilled 4% paraformaldehyde (pH 7.4 in 0.1 M PBS) by peristaltic pump. Mouse pups (<5 g) were anesthetized using a combination of hypothermia (place on ice) and ketamine/xylazine (3:1; I.M.) before transcardial perfusion with PBS (2 mL, room temp) followed by 4% paraformaldehyde. Brainstem sections were removed and allowed to post-fix for up to 24 h in 4% paraformaldehyde.

Tissue sections (75 μm thick coronal slices) were collected using a Zeiss VT1000S vibratome. Slices were permeabilized by treating with a mixture of 0.5% Triton-X/10% normal horse serum/PBS for 2 h. Sections were then transferred to a 0.1% Triton-X/10% normal horse serum/PBS mixture with the primary antibody (1:100, goat anti-mouse Phox2b; 1:100, mouse anti-mouse somatostatin; and/or 1:500 rabbit-anti Lucifer yellow) for 16 h. The tissue was then washed three times in 0.1% Triton-X/10% FBS/PBS solution; the secondary antibody (1:500, donkey anti-goat Cy3; 1:500 donkey anti-mouse AlexaFluor 647; and/or 1:500 donkey anti-rabbit AlexaFluor 488) was incubated with the tissue after the third wash for 2 h. Tissue sections were then washed three times in PBS before mounting on pre-cleaned glass slides with Prolong Gold with DAPI (ThermoFisher).

*Imaging and cell counting.* Slices containing the region of interest was imaged over a depth of 75 μm on the Z plane using 10× and 20× objectives and a Keyence BZ-X700 microscope. For caudal nucleus tractus solitarius (cNTS) slices, the central canal was centered and focused for equal representation the region in both hemispheres. For locus coeruleus (LC) and retrotrapezoid nucleus (RTN) slices, only one side was imaged at a time. For RTN slices, the entire RTN was capture along the ventral surface, ensuring that the trigeminal ridge was visible as well as part of the facial nucleus, as evidenced by large Phox2b-IR cells. For pre-BötC slices, the nucleus ambiguous (NA) was identified by Phox2b immunolabeling; the area ~75 μm ventromedial to the NA was imaged for Sst-IR pre-BötC cells. Each image was Z compressed with full focus and max intensity using the Keyence BZ-X700 Image Analyzer software. Parafacial cells were identified with co-localization of Phox2b-IR and DAPI in 75 μm × 75 μm blocks along the ventral surface for 600 μm medial to the trigeminal nucleus and from the rostral nucleus ambiguous (caudal border) to the facial nerve tracts (rostral border). The number of Phox2b-IR cells in the cNTS and LC was counted by hand in ImageJ for each 75 μm slice and quantified manually across slices. For anterograde tracer imaging, 40× images were taken on a Leica SP8 confocal microscope and manually analyzed in ImageJ for localization of eYFP puncta along with DAPI.

**Single cell isolation**. A filtered single cell population of parafacial cells was obtained as previously described[51]. In brief, animals were euthanized under anesthesia (3% isoflurane) and brainstem slices were prepared using a vibratome in ice cold, high sucrose slicing solution containing (in mM): 87 NaCl, 75 sucrose, 25 glucose, 25 NaHCO₃, 1.25 NaH₂PO₄, 2.5 KCl, 7.5 MgCl₂, 0.5 mM CaCl₂, and 5 L-ascorbic acid. Slicing solution was equilibrated with a 5% CO₂−95% O₂ gas mixture. Transverse slices (300 μm thick) were prepared and transferred to a glass Petri dish containing ice cold dissociation solution composed of (in mM): 185 sucrose, 10 glucose, 30 Na₂SO₄, 2 K₂SO₄, 10 HEPES, 0.5 CaCl₂, 6 MgCl₂, 5 L-ascorbic acid, pH 7.4, 320 mOsm. Using a plastic transfer pipette and scalpel (15 blade), the medial and lateral parafacial region was isolated and manually separated into sterile microcentrifuge tubes. The tissue chunks were then warmed to 34 °C for 10 min followed by trituration using a 25 G and 30 G needle sequentially, attached to a 3 mL syringe. Samples were triturated for an average of 5 min. The samples were placed back on ice and filtered through a 30-micron filter (Miltenyi Biotech) into round bottom polystyrene tube for scRNAseq or FACS.

**Single cell RNA sequencing and analysis**. Four libraries from two batches, all generated at The Jackson Laboratory for Genomic Medicine, were analyzed in this study; three batches were consistent of control mice (age and sex matched) while the fourth consisted of $Tcf4^{tr/+}$ mice. The first batch of two libraries was generated from a previous study and downloaded from GEO (accession GSE125065). FASTQ files from all four libraries were aligned to the GRCm38 genome annotated with Gencode vM23 annotations (mm10-2020A reference, 10× Genomics) using 10× Genomics CellRanger 4.0.0.

After cell-by-gene count matrix generation, all data were analyzed with Scanpy 1.6.1 and AnnData 0.7.5[53]. Scrublet[54] was used to assign putative doublet scores to all cells in each raw count matrix. The following quality control metrics were computed and used to exclude cells from each count matrix individually: UMIs per cell less than 1500 or greater than 25,000; genes per cell <1000 or greater than 8000; percentage of counts assigned to mitochondrial genes greater than 15%; more than 20 counts of hemoglobin genes per cell. These criteria resulted in filtered 20,909 cells across the four libraries.

Filtered matrices were then concatenated, normalized by per-cell library size (scaled to the dataset median UMIs per cell), and log transformed. The 2000 most highly variable genes were used to perform compute principal components, which were subsequently corrected for differences in 10× chemistry using Harmony (theta = 2)[55]. The first 20 corrected PCs were used to compute a $k = 15$ nearest neighbor graph using cosine distance and embedded with UMAP[56]. Lieden community detection[57] was used to label global populations. Populations expressing typical marker genes of multiple distinct cell types and those with high median Scrublet doublet scores were excluded from downstream analysis.

Barcodes of neurons, determined by distinct clustering and expression of *Snap25*, *Tubb3*, and *Elavl2* were isolated and used to subset the raw, merged dataset. The general analysis strategy outlined above, including re-normalization, highly variable gene selection, and batch correction, was used to re-embed and cluster the neurons.

**Florescence-activated cell sorting (FACS) and qRT-PCR**. A single cell suspension of TdT-positive parafacial cells or TdT-positive astrocytes were sorted on a BD FACSAria II Cell Sorter (UConn COR²E Facility, Storrs, CT) equipped with 407, 488, and 607 nm excitation lasers. Five minutes before sorting, 5 μL of 100 ng/mL DAPI was added to each sample. Cells were gated based on scatter (forward and side), for singlets, and for absence of DAPI (Supplementary Fig. 10). Finally, cells were gated to TdTomato and sorted by 4-way purity into a sterile 96-well plate containing 5 μL of sterile PBS per sample. Thousand five hundred cells were sorted per sample in an experiment and were processed immediately following FACS.

*Pooled cell qRT-PCR*. A lysis reaction followed by reverse transcription was performed using the kit Taqman Gene Expression Cells-to-CT Kit (ThermoFisher) with "Lysis Solution" followed by the "Stop Solution" at room temperature, and then a reverse transcription with the "RT Buffer", "RT Enzyme Mix", and lysed RNA at 37 °C for an hour. Following reverse transcription, cDNA was pre-amplified by adding 2 μL of cDNA from each sample to 8 μL of preamp master mix [5 μL TaKaRa premix Taq polymerase (Clontech), 2.5 μL 0.2× Taqman pooled probe, 0.5 μL H₂O] and thermocycled at 95 °C for 3 min, 55 °C for 2 min, 72 °C for 2 min, then 95 °C for 15 s, 60 °C for 2 min, 72 °C for 2 min for 16 cycles, and then a final 10 °C hold. Amplified cDNA was then diluted 2:100 in RNase free H₂O. Each qPCR assay contained the following reagents: 0.5 μL 20× Taqman probe, 2.5 μL RNase free H₂O, 5 μL Fast Advanced Master Mix (ThermoFisher), and 2 μL diluted pre-amplified cDNA. qPCR reactions were performed in triplicate for each Taqman assay of interest on a QuantStudio 3 Real Time PCR Machine (ThermoFisher).

*qRT-PCR data analysis*. Three technical replicates were averaged to determine a raw Ct value for each Taqman assay in QuantStudio Design & Analysis Software v1.5.1. We used specific markers as positive and negative controls, depending on which cell population we sorted for. For neurons, astrocytes (*Aldh1l1*) and GABAergic neurons (*Slc32a1*) were used as negative cell controls (no raw Ct after

40 cycles) and a pan neuron marker (*Rbfox3*) as well as *Slc17a6* for glutamatergic neurons as positive controls. However, for astrocytes, *Rbfox3* (for neurons), *Olig1* (for oligodendrocytes), and *Cx3cr1* (for microglia) were used as negative controls. Water was also used as a no template control for all reactions. *Gapdh* was used as a sample dependent internal control (Raw Ct value 16–20). Raw Ct values are indicated in text where appropriate.

**Brain slice preparation and electrophysiology**. Slices containing the RTN were prepared as previously described[39]. In short, *Tcf4* mice were anesthetized by administration of ketamine (375 mg/kg, I.P.) and xylazine (25 mg/kg; I.P.) and rapidly decapitated; brainstems were removed and transverse brainstem slices (250–300 μm) were cut using a microslicer (DSK 1500E; Dosaka) in ice-cold substituted Ringer solution containing the following (in mM): 260 sucrose, 3 KCl, 5 MgCl₂, 1 CaCl₂, 1.25 NaH₂PO₄, 26 NaHCO₃, 10 glucose, and 1 kynurenic acid. Slices were incubated for 30 min at 37 °C and subsequently at room temperature in a normal Ringer's solution containing (in mM): 130 NaCl, 3 KCl, 2 MgCl₂, 2 CaCl₂, 1.25 NaH₂PO₄, 26 NaHCO₃, and 10 glucose. Both substituted and normal Ringer's solutions were bubbled with 95% O₂ and 5% CO₂ (pH = 7.30).

Individual slices containing the RTN were transferred to a recording chamber mounted on a fixed-stage microscope (Zeiss Axioskop FS) and perfused continuously (~2 mL/min) with a bath solution containing (in mM): 140 NaCl, 3 KCl, 2 MgCl₂, 2 CaCl₂, 10 HEPES, 10 glucose (equilibrated with 5% CO₂; pH = 7.3). All recordings were made with an Axopatch 200B patch-clamp amplifier, digitized with a Digidata 1322AA/D converter and recorded using ClampEx 11.0.3 software. Recordings were obtained at room temperature (~22 °C) with patch electrodes pulled from borosilicate glass capillaries (Harvard Apparatus, Molliston, MA) on a two-stage puller (P-97; Sutter Instrument, Novato, CA) to a DC resistance of 5–7 MΩ when filled with pipette solution. Electrode tips were coated with Sylgard 184 (Dow Corning, Midland, MI). For both configurations below, only one cell was recorded per slice.

*Cellular firing behavior*. Firing activity was measured in the cell-attached (seal resistance > 1 GΩ) voltage clamp (Vhold −60 mV) configuration using a pipette solution containing (in mM): 125 K-gluconate, 10 HEPES, 4 Mg-ATP, 3 Na-GTP, 1 EGTA, 10 Na-phosphocreatine, 0.2% Lucifer yellow (pH 7.30). Firing rate histograms were generated by integrating action potential discharge in 10 s bins using Spike 5.0 software. For each experiment, we introduce 10% CO₂ for at least 5 min or when a plateau of firing activity is achieved for at least 2 min.

*Synaptic recordings*. Spontaneous synaptic currents were characterized in whole-cell voltage-clamp mode using a Cs-based pipette solution containing the following: 135 mM CsCH₃SO₃, 10 mM HEPES, 1 mM EGTA, 1 mM MgCl2, 3.2 mM TEA-Cl, 5 mM Na-phosphocreatine, 4 mM MgATP, and 0.3 mM NaGTP. To record spontaneous IPSCs (sIPSCs), cells were held at the reversal potential for AMPA-mediated excitatory synaptic currents (sEPSCs; Ihold = 0 mV). EPSCs were recorded at a holding potential of the measured IPSC reversal of −60 mV. Spontaneous EPSCs and IPSCs were analyzed using the Mini Analysis Program (Synaptosoft), events were identified based on amplitude (minimum 5 pA) and characteristic kinetics (fast rising phase followed by a slow decay). Each automatically detected event was also visually inspected to exclude obvious false responses. All whole-cell recordings had an access resistance (Ra) < 20 MΩ, recordings were discarded if Ra varied 10% during an experiment, and capacitance and Ra compensation (70%) were used to minimize voltage errors. A liquid junction potential of –10 mV (KCH₃SO₃) or +11 mV (CsCH₃SO₃) was corrected off-line.

**Unrestrained whole-body plethysmography**

*Adult chamber*. Respiratory activity was measured by whole-body plethysmograph (DSI/Buxco, St. Paul, MN) using a small animal chamber maintained at room temperature and ventilated at 1.16 L/min. Chamber temperature and humidity was continuously monitored and used to correct tidal volume breath by breath basis. Mice were individually placed into the chamber and allowed 1 h to acclimate prior to the start of an experiment. Respiratory activity was recorded using Ponemah 5.32 software (DSI) for a period of 15–30 min in room air followed by exposure to graded increases in CO₂ from 0 to 7% CO₂ (balance O₂). We also measured the ventilatory response to CO₂ ~1.5 h following systemic administration of saline (30 μL), PF-04531083 (40 mg/kg) or PF-06305591 (2 mg/kg). In separate experiments, we characterized the ventilatory response to hypoxic stimuli: 10% O₂ (balance N₂).

*Mouse pup chamber*. A pup whole-body plethysmograph chamber was used to measure respiratory activity in mouse pups (<15 days of age) (Buxco/DSI, St. Paul, MN). Pups are placed on a warming stage with the plethysmography chamber that is maintained at 31.5 °C to minimize loss of body temperature during an experiment, while at the same time ensuring a robust signal to noise ratio. Chamber temperature and humidity was monitored and used to correct tidal volume on a breath by breath basis. Pup ventilatory responses to high CO₂ and low O₂ were characterized as described above for adult animals. We also characterized the

ventilatory response to $CO_2$ in a separate cohort of pups that received systemic injection of saline (10 µL), PF-04531083 (40 mg/kg) or PF-06305591 (2 mg/kg).

*Analysis.* Plethysmography experiments were video recorded and sections of data containing behavior artifacts were excluded from analysis. A 20 min section of data was use for assessment of baseline respiratory pattern and a 1 min section of data was used to determine respiratory frequency, tidal volume, and minute ventilation; data was binned at 1 s for graphical display. An apnea was conservatively defined as three or more missed breaths and sighs were identified based on their characteristic large amplitude ($2\times V_T$) and followed by a post-sigh apnea. Parameters of interest include: respiratory frequency (breaths/minute), tidal volume ($V_T$, measured in mL; normalized to body weight and corrected to account for chamber temperature, humidity, and atmospheric pressure), and minute ventilation ($V_E$, mL/min/g). A 20 s period of relative quiescence after 4–5 min of exposure to each condition was selected for analysis. All experiments were performed between 9 a.m. and 6 p.m. to minimize potential circadian effects.

**Open field assay.** Locomotor activity and anxiety was assessed using Noldus PhenoTyper cages as previously shown[58]. Each cage is outfitted with two sets of cameras; one on the ceiling that faces the platform (35 cm × 35 cm) and another pointed on the side of the cage. Mice were acclimated to the experimentation room in their home cages for at least 1 h. During acclimation, the Noldus EthoVision software was set-up to track movement for 30 total minutes. After acclimation, mice were placed in the center of the open field, opaque Plexiglas was placed on all four sides of the cage to obscure any visual cues, and the trial was started in the EthoVision software. After 30 min, the trial ended and mice were placed back into their home cages. A separate cohort of mice received systemic (I.P.) injections of PF-04531083 (40 mg/kg) or PF-06305591 (2 mg/kg) 1.5 h prior to placement in the novel open field. Noldus EthoVision software was used to determine distance traveled, time spent in center, and frequency of going to center.

**Diaphragm and abdominal electromyography (EMG) recording.** Adult mice of each genotype were anesthetized with isoflurane (1.5%) and positioned on a heating pad to maintain body temperature. The skin covering the lateral intercostal muscle to the rectus abdominus muscle was resected and bathed with sterile saline. Silver wire electrodes were inserted into the diaphragm and the lateral portion of the rectus abdominus; these were connected to an A-M Systems 1700 differential AC amplifier (gain 10k, no filter). Raw diaphragm ($Dia_{EMG}$) and abdominal ($Abd_{EMG}$) activity was recorded using PowerLab 26T and LabChart 8 (ADInstruments). Raw diaphragm and abdominal EMG recordings included an electrocardiogram (ECG) artifact that was removed through thresholding in LabChart. Integrated diaphragm ($\int Dia_{EMG}$) and abdominal ($\int Abd_{EMG}$) muscle activities were obtained after removal of ECG contaminating signal and smoothened using a triangular (Barlett) window of 101 samples. Amplitude and frequency of each EMG signal was evaluated from a 20 s section of data acquired during exposure to 0, 5, and 7% $CO_2$ before and 1.5 h after systemic application of saline (30 µL) or PF-04531083 (40 mg/kg).

**Electrocorticography (ECoG) and electromyography (EMG) recordings**
*Placement.* Adult $Tcf4^{tr/+}$ mice (>20 grams) were anesthetized with an induction dose of 3% isoflurane and placed into a sterile field. A large skin incision was made parallel to the eyes, above the skull, and past the trapezius muscles; a large left sided skin flap was created using blunt dissecting scissors and filled with 1 mL saline. After, an HD-X02 transmitter (Data Sciences International; DSI) was placed into a dorsal flank skin flap. One pair of biopotential leads were placed into the left trapezius muscle of the animal. The other pair of biopotential leads were placed just under the skull left of bregma and the other to the right of the sagittal suture in between bregma and lambda. These leads were secured with dental cement (Henry Schein Dental). After surgery mice were placed on a headed pad until conscious. Meloxicam was administered 0, 24, and 48 h postoperatively. Mice were given at least seven days to recover before experimentation.

*ECoG and EMG recording and analysis.* Animals were allowed to stay in their home cages for the entirety of the acquisition period. Wireless telemetry units were turned on and each mouse's home cage was placed on the appropriate receiver unit. Raw ECoG and EMG traces were recorded using Ponemah v5.32 (DSI). After at least 24 h of continuous monitoring, the recording was stopped and telemetry unit was turned off.

Individual recordings were loaded into NeuroScore (DSI; version 3.3.1). Raw ECoG and EMG was scored in 10 s epochs in the following categories: wake, non-REM sleep, REM sleep, and seizure. To detect seizure-like activity we continuously monitored ECoG and EMG activity 24 h. For presentation, we selected a representative 1 h section of data recorded at the same time of day. NeuroScore also calculated the periodogram power spectrum bands for the selected hour of EEG recordings based on the following distinctions: delta from 0.5 to 4 Hz, theta from 4 to 8 Hz, alpha from 8 to 12 Hz, sigma from 12 to 16 Hz, and beta from 16 to 24 Hz.

**Viral injections into the medial parafacial region (RTN).** Adult mice (>20 g) were anesthetized with 3% isoflurane. The right cheek of the animal was shaved, and an incision was made to expose the right marginal mandibular branch of the facial nerve. The animals were then placed in a stereotaxic frame and a bipolar stimulating electrode was placed directly adjacent to the nerve. Animals were maintained on 1.5% isoflurane for the remainder of the surgery. An incision was made to expose the skull and two 1.5 mm holes were drilled left and right of the posterior fontanelle, caudal of the lambdoidal suture. The facial nerve was stimulated using a bipolar stimulating electrode to evoke antidromic field potentials within the facial motor nucleus. In this way, the facial nucleus on the right side of the animal was mapped in the X, Y, and Z direction using a quartz recording electrode.

The viral vector (AAV2-Ef1α-DIO-hChR2(H134R)-EYFP or AAV2-hSyn-DIO-eGFP-mScn10a-shRNAmir) was loaded into a borosilicate glass pipette (I.D. 1.2 mm) and positioned in a Nanoject III system (Drummond Scientific). Virus was injected at least −0.02 mm ventral to the Z-coordinates of the facial nucleus, to ensure injection into the RTN region. These same coordinates were used for the left side of the animal. In all mice, incisions were closed with nylon sutures and surgical cyanoacrylate adhesive. Mice were placed on a heated pad until consciousness was regained. Meloxicam was administered 24 and 48 h postoperatively.

*Viral construct.* To select for the shRNAmir sequence that yielded the best knockdown, four different clone sequences for mScn10a shRNA were inserted into vectors and overexpressed in heterologous cell lines. After, clones were screened using qPCR for total knockdown of mRNA particles; clone #3 produced a 77% knockdown of the mouse *Scn10a* gene with the following sequence: 5′-GCTGAA GACTGTGAGGATGGTCACGGTTTTGGCCACTGACTGACCGTGACCACTC ACAGTCTTCAG-3′ and was used in the final viral preparation.

**Reporting summary.** Further information on experimental design is available in the Nature Research Reporting Summary linked to this paper.

## Data availability
Data supporting the findings in this study are included within the Supplementary Material and available from the corresponding author on request. The source data relevant to Figs. 1–9 and Supplementary Figs. 3–9 are provided as a Source Data file. Single-cell RNA sequencing data has been deposited in the GEO database under accession code GSE174417 and is available publicly without restriction. Source data are provided with this paper.

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

## Acknowledgements

We thank Huda Zoghbi for encouraging us to pursue this research topic. We thank Ji-Young Lee and Yong-Ki Park for help with metabolic analysis, Alex Jackson for guidance with the open field assay, and Thiago Moreira for assistance using electromyography to measure respiratory muscle activity. We also thank William Flynn and Paul Robson for single cell RNA sequencing assistance. This work was supported by the following National Institutes of Health Grants: HL104101 (D.K.M.), HL137094 (D.K.M.), NS099887 (D.K.M.), MH104593 (B.J.M.) and MH110487 (B.J.M.) and F31HL142227 (C.M.C.). This work was also supported in part by funding from the Pitt Hopkins Research Foundation (B.J.M.).

## Author contributions

C.M.C.: Designed experiments, Generated data, analyzed results, edited the manuscript, approved final manuscript, S.J.: generated data and edited the manuscript, approved final manuscript, B.J.M.: Designed experiments and edited the manuscript, approved final manuscript, D.K.M.: Designed experiments, drafted the manuscript, and approved final manuscript.

## Competing interests

The authors declare no competing interests.
