## [Peer Review File · Nature Communications]

Disordered breathing in a Pitt-Hopkins syndrome model involves Phox2b-expressing parafacial neurons and aberrant Nav1.8 expressionREVIEWER COMMENTS

Reviewer #1 (Remarks to the Author):

Pitt-Hopkins syndrome (PTHS) is a rare condition and only ca. 500 patients have been reported worldwide. Symptoms include intellectual disability, impaired speech recurrent seizures, and breathing abnormalities. The latter involve, bouts of hyperventilation (without a clear trigger) that can last ca. 5 minutes followed by apnea. The breathing disorder is state-dependent and manifest during wakefulness during early infancy and adolescence.

The present study by Cleary et al., investigates the origins of breathing disorders in a mouse model of PTHS (Tcf4tr/+) which is based on a Tcf4truncation. The working hypothesis of the authors suggested that periodic breathing as main respiratory symptom in PTHS is linked to dysregulation of chemoreception and blood gas homeostasis.

Strength of the study:

The study presents substantial amount of data including in vitro electrophysiology, in vivo plethysmography, the analysis of inspiratory diaphragm and expiratory muscle activity, and immunohistochemistry. The data presented illustrate that impaired chemosensitivity in Tcf4tr/+ mice is directly linked to selective cell loss of PHOX2B expressing neurons of pFRG/RTN cell group in the medulla oblongata. Important findings of the study also include reduced connectivity of RTN chemosensors with the inspiratory rhythm generating pre-BotC. The absence of abdominal EMG activity during exposure with 7% CO₂ is another important verification of impaired response to hypercapnia in Tcf4tr/+ mice. Overall, the study provides strong evidence for the working hypothesis of the authors that impaired central chemosensitivity is the primary source of breathing abnormalities in PTHS.

Weakness of the study:

The hyperventilation-apnea symptom of PTHS is linked to wakefulness, while the key reference (10) for the working hypothesis refers to changes in chemoreception during sleep? A seizure phenotype of Tcf4tr/+ mice is not reported. The clinically relevant respiratory phenotype of periodic breathing hyperventilation followed by apnea appears to be not present in Tcf4tr/+ (see figures). Instead the study identifies prolonged post sighs apneas, instead of the clinical symptom of periodic breathing. Longer recording of 20 mins respiratory activity showing a clear periodic breathing phenotype needs to be illustrated.

The proposed therapy apparently triggered pathological prolongation of phrenic nerve discharge and combined with significant shortening of expiratory duration (see supply Fig. 4a, 0% and 7% CO₂), this potential adverse side-effect is neither analyzed nor discussed. Moreover, treatment with PF-06305591, a BBB permeable Nav1.8 blocker, only partially (ca.50%, Fig. 3B) restored CO₂ sensitivity, thus a proclamation that a Nav1.8 blocker is a high priority target with therapeutic potential in PTHS, seems to be overstated. Plethysmographic analysis was performed between 9 a.m and 6 pm – however it is not mentioned whether the mice were kept on a reversed 12 hr light dark cycle. Since PTHS breathing disorder are reported to occur during wakefulness, the plethysmography needs to be conducted during the dark cycle to match breathing abnormalities of PTHS.

The breathing phenotype of Tcf4tr/+ could have been compared and discussed to the extensively investigated breathing phenotypes of Mecp2 deficient mice as another mouse model for autism spectrum disorder with severe and similar breathing disorders seen in PHTS.

Reviewer #2 (Remarks to the Author):

In the paper entitled “Disordered breathing in Pitt-Hopkins syndrome involves disruption of parafacial respiratory neurons and aberrant expression of Nav1.8.” the authors aim to elucidate the mechanisms contributing to respiratory deficits associated with the Pitt-Hopkins syndrome, together with identifying potential candidates to be targeted as therapeutic treatments. PTHS is a rare genetic disorder caused by haploinsufficiency of a gene encoding the transcription factor 4 (Tcf4). In their study, the authors used a mouse model that recapitulates in a certain fashion respiratory deficits observed in humans. For instance, the mutant mice exhibit a reduced occurrence of sigh events associated with longer post-sigh apneas. In addition, the animals bearing the genetic deletion do not respond properly to elevated levels of CO₂. They showed that these anomalies are to be related to a selective loss of some neurons constituting the retrotrapezoid group (RTN) known to be involved in central chemoception and some neurons involved in active expiration (pFL neurons), both types of neurons being activated during CO₂ challenges. Moreover, they provide evidences for a potential therapeutic way of treating PTHS deficits by acting on central Nav1.8 channels, as treatment with a Nav1.8 blocker alleviates respiratory deficits at the cellular and behavioral levels.

While the paper is nicely written and provides convincing data, I have a few concerns that should be addressed to clarify some points:

- The mouse model used here for PTHS exhibits breathing anomalies characterized by a lower frequency of sighs that are associated with longer post-sigh apneas and a diminished response to elevated CO₂. But, the PTHS is generally associated to hyperventilation with intermittent apneas, without any mention of dysfunction in central chemoception. So I wonder whether considering that this mouse replicates respiratory deficits seen in humans is fully correct. Indeed sigh cannot be considered as hyperventilation, and post-sigh apneas have underlying mechanisms that are very different than those involved in central and even obstructive apneas. Also the authors state that WT and mutant mice exhibit the same occurrence of spontaneous apneic events. This is not mimicking the human phenotype.

- It remains unclear what is the status of the pFL neurons. Indeed in some places it is: an unidentified group of neurons located in the adjacent parafacial lateral region (pFL), in another one it is parafacial Phox2b⁺ neurons (putative expiratory pFL neurons) and then they are neurons without cell-type specific markers but that could be included in the cluster revealed in the single cell RNA sequencing experiments. Finally, they end up in the discussion as being: The lateral parafacial region is thought to regulate active expiration; however, it is not clear whether pFL neurons express Phox2b. The authors must make their position clear on the identity of those cells. This is extremely important as everything is based on the fact that the neurons of interest here are Phox2b⁺ and more importantly must be specified by Tcf4 probably interacting by Atoh1 and that one of the major finding is a default in triggering active expiration in mutant animals.

- Related to the previous point, one major finding of the study is that the expiratory response to augmented CO₂ is affected in the mutant. This could be related to the fact that most of the pFL neurons (if we agree they are) are missing in the mutant. Therefore, to my point of view the results illustrated in supplemental Figure 4 should be better placed as a full Figure and not as a supplemental material.

- In the same vein I don't understand why data on the effects of the Nav1.8 blocker on baseline activity of RTN chemoreceptors are presented in a supplemental figure (suppl 8), whereas these results are important to validate the effect of the blocker at the cellular level.

- It has been shown in the literature that Nbm⁺ neurons located in the parafacial region modulate sigh activity generated in the preBötC network. Here the authors describe first a decrease in sigh frequency in the mutant and a loss of cells in the parafacial region (possibly including Nbm⁺ ones). Wouldn't it be reasonable to link these two observations?

- In Figure 2Bii and suppl 5B: I personally find this type of representation difficult to read and not

intuitive nor visual.

- Projections from RTN neurons towards preBötC neurons: is there an overall reduction in the number of projections. This would be expected as the number of RTN neurons is decreased by 21%. Thus, not only the projections are targeting different cells but also in a fewer number. If this is true wouldn't we expect a lower breathing frequency (that is not observed)?

- Figure 3: the samples of RTN neuron firing in the Tcf4 mutant do not seem to fit with the curve above, the graphs in C and B and the text. The same holds true with traces in E and graphs in F and G. The mutants seem to have more EPSC than the WT. Could you provide better samples?

-Isn't it reductive to argue that the decreased number of excitatory inputs to the RTN neurons could be explained by the fact that they are less numerous and thus less inter-connected? It is obvious that RTN receive inputs from other places.

Reviewer #3 (Remarks to the Author):

The authors constructed a mouse model of TCF4 and observed respiratory problems similar to those of patients with PTHS syndrome, observed impaired ability to regulate respiration in response to CO₂ in TCF4tr/+ mice, and found that Phox2b+ parafacial neurons, including RTN chemoreceptors, were depleted in Tcf4tr/+ mice, and experimentally found that the remaining RTN chemoreceptors in Tcf4tr/+ mice showed reduced CO₂/H⁺ responsiveness and altered connectivity with pre-BötC. Rescue experiments validated that central Nav1.8 channels can be pharmacologically targeted to improve chemoreceptor function at the cellular and behavioral levels in TCF4tr/+ mice, they found that Nav1.8 could be a priority target with therapeutic potential for PTHS syndrome. While some of the data presented are exciting and support some of their hypotheses, but also have some major weaknesses, which need to be addressed to justify the conclusions the authors propose.

Questions:

1. Patients with PTHS syndrome exhibit abnormal breathing patterns, developmental delays, impaired speech, and repetitive non-functional movements, which are consistent with autism spectrum disorders. The TCF4tr/+ mice constructed in this paper are considered to be behaviorally consistent with the PTHS syndrome model only by performing the absentee field experiment, and it is suggested to add other relevant behavioral experiments, such as the Marble Burying Test and the Ultrasonic Vocalization Test.

2. The legends in Fig.3A and B are not clearly labeled and confusing. Fig. 3E-I is not mentioned accordingly in the manuscript.

3. Many elaborations in the manuscript are not clear enough. PF-04531083 is mentioned first in Supplementary Figure 4A without any explanation of its role, and in the last part of manuscript that PF-04531083 is mentioned as a blocker of Nav1.8. The order of the supplementary material needs to be better organized.

4. The detection of reduced CO₂/H⁺ sensitivity at the synaptic level of chemosensitive RTN neurons in slices from TCF4tr/+ mice, is synaptic morphology also altered?

5. TCF4 is required for the differentiation of the Atoh1 group of cell subpopulations, and Atoh1 is required for the development of Phox2b-expressing parafacial neurons, single-cell sequencing and other experiments suggest interactions between TCF4 and Atoh1 contribute to cell-type specific deficits in PTHS. The authors should perform ISH and/or IF for TCF4 with Atoh1 in TCF4+/+ and TCF4tr/+ to verify its expression and effect.

6. Atoh1 affects neonatal respiratory efficacy during RTN neuronal development ((Huang et al,2012,

van der Heijden ME et al,2018),) and the authors found that deletion of TCF4 haplogroups also affects neonatal respiratory efficacy, as previously reported (Adriano Flora et al,2007), but how TCF4 interacts with Atoh1 in TCF4tr/+ mice to affect the respiratory pattern of mice was not elaborated in this manuscript.

7. It is mentioned in the method that adult mice were individually housed on a 12:12 light dark cycle in plastic cages with a running wheel for one week before experimentation. Will having a roller in the cages increase the activity of the mice and will it affect the measurement of respiratory activity of the mice?

8. Supplementary Figure 9 is mentioned in the manuscript, but Supplementary Figure 9 was not found.

Response to Reviewer Concerns:

Summary:

We thank the Reviewers for their support and thoughtful suggestions. We have addressed all reviewer concerns and modified the text and figures accordingly. In particular, we performed a more detailed characterization of baseline breathing and now confirm that *Tcf4^{tr/+}* mice exhibit unstable periodic breathing under room air conditions that is reminiscent of PTHS patients. However, contrary to PTHS patients breathing problems in *Tcf4^{tr/+}* mice were not worse during wakefulness; respiratory phenotype of *Tcf4^{tr/+}* mice was similar during both light/inactive and dark/active conditions. We also included new data showing that RTN chemoreceptor specific knockdown of *Scn10a* improved respiratory activity in *Tcf4^{tr/+}* mice. These results are consistent with our main conclusion that disordered breathing in Pitt Hopkins syndrome involves disruption of parafacial respiratory neurons and aberrant expression of Nav1.8.

Please see below our point-by-point responses to noted concerns.

Reviewer #1 Comments:

*Pitt-Hopkins syndrome (PTHS) is a rare condition and only ca. 500 patients have been reported worldwide. Symptoms include intellectual disability, impaired speech recurrent seizures, and breathing abnormalities. The latter involve, bouts of hyperventilation (without a clear trigger) that can last ca. 5 minutes followed by apnea. The breathing disorder is state-dependent and manifest during wakefulness during early infancy and adolescence. The present study by Cleary et al., investigates the origins of breathing disorders in a mouse model of PTHS (*Tcf4tr/+*) which is based on a *Tcf4truncation*. The working hypothesis of the authors suggested that periodic breathing as main respiratory symptom in PTHS is linked to dysregulation of chemoreception and blood gas homeostasis.*

Strength of the study: *The study presents substantial amount of data including in vitro electrophysiology, in vivo plethysmography, the analysis of inspiratory diaphragm and expiratory muscle activity, and immunohistochemistry. The data presented illustrate that impaired chemosensitivity in *Tcf4tr/+* mice is directly linked to selective cell loss of PHOX2B expressing neurons of pFRG/RTN cell group in the medulla oblongata. Important findings of the study also include reduced connectivity of RTN chemosensors with the inspiratory rhythm generating pre-BotC. The absence of abdominal EMG activity during exposure with 7% CO₂ is another important verification of impaired response to hypercapnia in *Tcf4tr/+* mice. Overall, the study provides strong evidence for the working hypothesis of the authors that impaired central chemosensitivity is the primary source of breathing abnormalities in PTHS.*

We thank the reviewer for highlighting these important aspects of this study.

Weakness of the study:

1a) The hyperventilation-apnea symptom of PTHS is linked to wakefulness, while the key reference (10) for the working hypothesis refers to changes in chemoreception during sleep?

Thanks for bringing this to our attention. The point we're trying to make is that disordered breathing in PTHS may involve compromised central chemoreception. Considering disordered breathing in PTHS is phenotypically similar to Rett syndrome (RTT), and since breathing

problems in RTT involve diminished central chemoreception, we have modified this section to read “For example, disordered breathing in PTHS is phenotypically similar to a related disorder known as Rett syndrome (RTT) (PMID: 22670143), and breathing problems in RTT involve disruption of central chemoreception (PMID: 26205541; PMID: 21307341). Also consistent with this possibility, acetazolamide...”

1b) A seizure phenotype of Tcf4tr/+ mice is not reported.

Good point. Approximately half of PTHS patients show seizure activity, over a developmental window similar to when breathing problems occur, that varies in severity from involuntary motor activity to tonic-clonic seizures (PMID: 27072915; PMID: 29318938). To determine whether *Tcf4^{tr/+}* exhibit seizure activity, we implanted radio telemetry units in *Tcf4^{tr/+}* and control mice to record electrocorticogram (ECoG) and electromyography (EMG) activity over a 24-hour period. Contrary to PTHS patients, we found that *Tcf4^{tr/+}* mice (age P60-64; n=6 mixed sex) did not exhibit overt seizure activity or large amplitude polyspike activity for the duration our recording. These results suggest *Tcf4^{tr/+}* mice at this developmental time-point do not show spontaneous seizure-like activity. These data have been added to the text and new supplemental Fig. 2, along with associated methods in the appropriate sections.

1c) The clinically relevant respiratory phenotype of periodic breathing hyperventilation followed by apnea appears to be not present in Tcf4tr/+ (see figures). Instead the study identifies prolonged post sighs apneas, instead of the clinical symptom of periodic breathing. Longer recording of 20 mins respiratory activity showing a clear periodic breathing phenotype needs to be illustrated.

Thank you for raising this important issue. As suggested, we analyzed an expanded section of data recorded under room air conditions (20 mins without behavioral artifact), and found that *Tcf4^{tr/+}* mice (n=6, mixed sex) show periodic breathing characterized by repeated cycles of waxing and waning of minute ventilation. This pattern of activity was not observed in *Tcf4^{tr/+}* mice (n=6, mixed sex) and resulted in unstable breathing as evidenced by a large increase in minute ventilation coefficient of variation (0.09 *Tcf4^{tr/+}* vs. 0.35 *Tcf4^{tr/+}*). These results show that *Tcf4^{tr/+}* exhibit a respiratory phenotype similar to PTHS patients. These new results are shown in Figs. 2A-C and the text has been modified accordingly.

2a) The proposed therapy apparently triggered pathological prolongation of phrenic nerve discharge and combined with significant shortening of expiratory duration (see supply Fig. 4a, 0% and 7% CO₂), this potential adverse side-effect is neither analyzed nor discussed.

Tcf4^{tr/+} mice that received the BBB permeable Nav1.8 blocker (PF-04531083) showed more stable breathing under baseline conditions (new Fig. 7A-B) and enhanced DiaEMG amplitude and frequency under high CO₂ conditions (new Fig. 8D-F); however, DiaEMG burst duration did not increase with this drug treatment even during exposure to 7% CO₂ (saline 0.3 ms vs PF-04531083 0.35 ms; p > 0.05). PF-04531083 also increased respiratory frequency in awake (not shown) and anesthetized (Fig. 8F) *Tcf4^{tr/+}* mice at 7% CO₂, consequently the time between breaths decreased. The inability of *Tcf4^{tr/+}* to generate active expiration may contribute to this frequency increase; however, since PF-04531083 also increased tidal volume we do not consider this a pathological response. We have expanded our discussion of these results.

2b) Moreover, treatment with PF-06305591, a BBB permeable Nav1.8 blocker, only partially (ca.50%, Fig. 3B) restored CO2 sensitivity, thus a proclamation that a Nav1.8 blocker is a high priority target with therapeutic potential in PTHS, seems to be overstated.

Previous work showed that Nav1.8 is upregulated in the brain of *Tcf4^{tr/+}* mice and pharmacological blockade of this channel improved *Tcf4^{tr/+}*-associated firing deficits in cortical pyramidal neurons (PMID: 28032012). Consistent with previous work, we show here that i) Nav1.8 transcript is upregulated in chemosensitive RTN neurons in *Tcf4^{w/+}* mice; ii) blocking Nav1.8 doubled CO₂/H⁺ sensitivity of RTN neurons in slices from *Tcf4^{w/+}* mice (new Fig. 6) and improved CO₂/H⁺-dependent respiratory output of anesthetized and awake *Tcf4^{w/+}* mice in vivo (new Figs. 7 and 8); and iii) Nav1.8 blockade also normalized hyper-locomotor activity and anxiety related behavior in *Tcf4^{w/+}* mice (new Fig. 9). These are the first evidence that selective blockade of Nav1.8 improved PTHS associated behaviors. Therefore, we do not believe it is an overstatement to identify Nav1.8 as a “high priority target with therapeutic potential in PTHS”.

2c) Plethysmographic analysis was performed between 9 a.m and 6 pm – however it is not mentioned whether the mice were kept on a reversed 12 hr light dark cycle. Since PTHS breathing disorder are reported to occur during wakefulness, the plethysmography needs to be conducted during the dark cycle to match breathing abnormalities of PTHS.

All experiments included in the original submission were performed using mice housed in a normal 12:12 light/dark cycle, and experiments were conducted during the light/inactive state. However, as noted by the reviewer, breathing problems in PTHS occur primarily during wakefulness (PMID: 29318938), so as suggested, we reassessed respiratory activity during the dark/active state in mice housed under reverse light-dark cycle conditions. We found that *Tcf4^{w/+}* mice exhibit a similar respiratory phenotype under both light and dark cycle conditions. Specifically, during the dark/active state *Tcf4^{w/+}* mice (N=6; P30-40, mixed sex) showed reduced sigh frequency and increase duration of post-sigh apneas under baseline conditions, and a blunted ventilatory response to CO₂. These results suggest that disordered breathing in *Tcf4^{w/+}* mice during wakefulness is similar between the dark/active and light/inactive periods. However, it should be noted that all experiments were performed on awake mice so it is unclear whether breathing problems persist in *Tcf4^{w/+}* mice during sleep. These new results have been included in the text.

*3) The breathing phenotype of *Tcf4^{tr/+}* could have been compared and discussed to the extensively investigated breathing phenotypes of *Mecp2* deficient mice as another mouse model for autism spectrum disorder with severe and similar breathing disorders seen in PHTS.*

Thank you for this suggestion. This is an interesting comparison particularly since both conditions show an unusual wake-dependence. We have added a paragraph to the discussion to expand on this point.

Reviewer

#2

Comments:

In the paper entitled “Disordered breathing in Pitt-Hopkins syndrome involves disruption of parafacial respiratory neurons and aberrant expression of Nav1.8.” the authors aim to elucidate

the mechanisms contributing to respiratory deficits associated with the Pitt-Hopkins syndrome, together with identifying potential candidates to be targeted as therapeutic treatments. PTHS is a rare genetic disorder caused by haploinsufficiency of a gene encoding the transcription factor 4 (Tcf4). In their study, the authors used a mouse model that recapitulates in a certain fashion respiratory deficits observed in humans. For instance, the mutant mice exhibit a reduced occurrence of sigh events associated with longer post-sigh apneas. In addition, the animals bearing the genetic deletion do not respond properly to elevated levels of CO₂. They showed that these anomalies are to be related to a selective loss of some neurons constituting the retrotrapezoid group (RTN) known to be involved in central chemoreception and some neurons involved in active expiration (pFL neurons), both types of neurons being activated during CO₂ challenges. Moreover, they provide evidences for a potential therapeutic way of treating PTHS deficits by acting on central Nav1.8 channels, as treatment with a Nav1.8 blocker alleviates respiratory deficits at the cellular and behavioral levels.

While the paper is nicely written and provides convincing data, I have a few concerns that should be addressed to clarify some points:

We thank the reviewer for their time and thoughtful suggestions.

1) The mouse model used here for PTHS exhibits breathing anomalies characterized by a lower frequency of sighs that are associated with longer post-sigh apneas and a diminished response to elevated CO₂. But, the PTHS is generally associated to hyperventilation with intermittent apneas, without any mention of dysfunction in central chemoreception. So I wonder whether considering that this mouse replicates respiratory deficits seen in humans is fully correct. Indeed sigh cannot be considered as hyperventilation, and post-sigh apneas have underlying mechanisms that are very different than those involved in central and even obstructive apneas. Also the authors state that WT and mutant mice exhibit the same occurrence of spontaneous apneic events. This is not mimicking the human phenotype.

Thank you for this comment. Note the CO₂ ventilatory response of PTHS patients has not been characterized, so we can only compare the respiratory phenotype of *Tcf4*^{tr/+} mice to PTHS patients under room air conditions. We agree that sighs are not a form of hyperventilation and post-sigh apneas are mechanistically dissimilar to central apnea. As such, our initial characterization of the baseline respiratory phenotype of *Tcf4*^{tr/+} mice did not fully reflect disordered breathing in PTHS patients. To address this further (as noted in response 1c above), we analyzed a longer section of breathing under room air conditions (20 mins), and in doing so, were able to find that *Tcf4*^{tr/+} mice (n=6 mice, mixed sex) show a periodic respiratory pattern consistent with that described for PTHS patients. For example, *Tcf4*^{tr/+} mice show repeated bouts of hyperventilation (high amplitude and high frequency) followed by periods of diminished respiratory activity. This waxing and waning of minute ventilation was not observed in *Tcf4*^{+/+} mice (n=6 mice, mixed sex) and resulted in unstable breathing as evidenced by a large increase in minute ventilation coefficient of variation (0.09 *Tcf4*^{t+/+} vs 0.35 *Tcf4*^{tr/+}). These new results show that *Tcf4*^{tr/+} exhibit a respiratory phenotype similar to PTHS patients. These new results are shown in Figs. 2A-C and the text has been modified accordingly.

2) *It remains unclear what is the status of the pFL neurons. Indeed in some places it is: an unidentified group of neurons located in the adjacent parafacial lateral region (pFL), in another one it is parafacial Phox2b+ neurons (putative expiratory pFL neurons) and then they are neurons without cell-type specific markers but that could be included in the cluster revealed in the single cell RNA sequencing experiments. Finally, they end up in the discussion as being: The lateral parafacial region is thought to regulate active expiration; however, it is not clear whether pFL neurons express Phox2b. The authors must make their position clear on the identity of those cells. This is extremely important as everything is based on the fact that the neurons of interest here are Phox2b+ and more importantly must be specified by Tcf4 probably interacting by Atoh1 and that one of the major finding is a default in triggering active expiration in mutant animals.*

Sorry for this confusion. Previous work suggests pFL neurons are glutamatergic but not Phox2b-immunoreactive (PMID: 28004411) and we make this clear throughout the text. However, results presented here suggest expiratory pFL neurons may be Phox2b+. Specifically, we show that glutamatergic Phox2b+ neurons are preferentially lost in the lateral parafacial region (new Fig. 4), and this coincided with loss of expiratory activity (new Fig. 3). Conversely, the proportion of glutamatergic Phox2b-negative parafacial was similar between genotypes. These results suggest expiratory pFL neurons are Phox2b+ but functionally distinct from chemosensitive RTN neurons that also express this transcription factor.

3) *Related to the previous point, one major finding of the study is that the expiratory response to augmented CO2 is affected in the mutant. This could be related to the fact that most of the pFL neurons (if we agree they are) are missing in the mutant. Therefore, to my point of view the results illustrated in supplemental Figure 4 should be better placed as a full Figure and not as a supplemental material.*

We agree, and have reorganized figures to show expiratory activity in control and *Tcf4*^{tr/+} mice under control conditions (new Fig. 3) and after injection of PF-045310803 (new Fig. 8).

4) *In the same vein I don't understand why data on the effects of the Nav1.8 blocker on baseline activity of RTN chemoreceptors are presented in a supplemental figure (suppl 8), whereas these results are important to validate the effect of the blocker at the cellular level.*

We agree. These results are now shown as main figure 6.

5) *It has been shown in the literature that Nbm+ neurons located in the parafacial region modulate sigh activity generated in the preBötC network. Here the authors describe first a decrease in sigh frequency in the mutant and a loss of cells in the parafacial region (possibly including Nmb+ ones). Wouldn't it be reasonable to link these two observations?*

Thanks for reminding us to make this point. We have added the following statement to the first paragraph of the discussion “Note that NMB expressing parafacial neurons, which includes Phox2b+ RTN neurons (PMID: 29066557), regulate sighing (PMID: 26855425), thus loss of this population in *Tcf4*^{tr/+} mice is consistent with diminished sigh activity”.

6) *In Figure 2Bii and suppl 5B: I personally find this type of representation difficult to read and not intuitive nor visual.*

We struggled with how best to represent these results. We decided on the plots in question because, in our opinion, they best represent changes in Phox2b+ distribution observed in our experimental model; loss of Phox2b+ cells in the lateral parafacial region and clumping of Phox2b+ cells in the medial parafacial region. Please note that raw cell count tabulations are also included in Source Data, thus providing individuals with the option of replotting these data in their preferred format.

7a) *Projections from RTN neurons towards preBötC neurons: is there an overall reduction in the number of projections. This would be expected as the number of RTN neurons is decreased by 21%. Thus, not only the projections are targeting different cells but also in a fewer number.*

Good point. We confirmed that the average number of pre-BötC neurons labeled (minimum of 5 eYFP puncta) with a Cre-dependent anterograde RTN tracer (AAV2-Ef1 α -DIO-hChR2(H134R)-EYFP) into the medial parafacial region decreased from 144 cells on average per control mouse to 112 cells average per *Tcf4*^{tr/+} mouse ($T_4=5.168$, $p=0.0067$). These results suggest that number of pre-BötC neurons receiving input from Phox2b+ RTN neurons decreased in *Tcf4*^{tr/+} mice compared to control. These new results are summarized in new Fig. 4F and discussed in the text.

7b) *If this is true wouldn't we expect a lower breathing frequency (that is not observed)?*

In theory, yes. However, this outcome likely depends on the level of RTN neural activity. For example, under baseline conditions, RTN neurons show relatively modest activity so it is reasonable to expect that a large number of RTN neurons must be lost before baseline breathing is affected. Conversely, when the activity of RTN neurons is stimulated by exposure to high CO₂, it is possible that loss of just a few RTN neurons can impact respiratory behavior. Consistent with this, *Tcf4*^{tr/+} mice show a hypoventilatory phenotype only under high CO₂ conditions.

8) *Figure 3: the samples of RTN neuron firing in the Tcf4 mutant do not seem to fit with the curve above, the graphs in C and B and the text. The same holds true with traces in E and graphs in F and G. The mutants seem to have more EPSC than the WT. Could you provide better samples?*

Thank you for bringing this to our attention. We have found more representative examples for cellular (new Fig. 5A,B) and synaptic data (new Fig. 5E).

9) *Isn't it reductive to argue that the decreased number of excitatory inputs to the RTN neurons could be explained by the fact that they are less numerous and thus less inter-connected? It is obvious that RTN receive inputs from other places.*

Please note that we are focusing on CO₂/H⁺-activated glutamatergic drive to the RTN. In the intact brain, there are numerous glutamatergic inputs to the RTN that could increase in proportion with respiratory activity. However, in the reduced brain slice, chemosensitive RTN neurons and possibly medullary raphe neurons are the most likely sources of CO₂/H⁺-dependent glutamatergic drive to RTN neurons (PMID: 34013884). Therefore, we think it is reasonable to

suggest loss of RTN neurons contributes to diminished CO₂/H⁺ dependent excitatory modulation of RTN neurons in *Tcf4^{tr/+}*. We have modified the text to make this clearer.

Reviewer #3 Comments:

The authors constructed a mouse model of TCF4 and observed respiratory problems similar to those of patients with PTHS syndrome, observed impaired ability to regulate respiration in response to CO₂ in TCF4^{tr/+} mice, and found that Phox2b⁺ parafacial neurons, including RTN chemoreceptors, were depleted in Tcf4^{tr/+} mice, and experimentally found that the remaining RTN chemoreceptors in Tcf4^{tr/+} mice showed reduced CO₂/H⁺ responsiveness and altered connectivity with pre-BötC. Rescue experiments validated that central Nav1.8 channels can be pharmacologically targeted to improve chemoreceptor function at the cellular and behavioral levels in TCF4^{tr/+} mice, they found that Nav1.8 could be a priority target with therapeutic potential for PTHS syndrome.

While some of the data presented are exciting and support some of their hypotheses, but also have some major weaknesses, which need to be addressed to justify the conclusions the authors propose.

We thank the reviewer for their time and thoughtful suggestions.

Questions:

1. Patients with PTHS syndrome exhibit abnormal breathing patterns, developmental delays, impaired speech, and repetitive non-functional movements, which are consistent with autism spectrum disorders. The TCF4^{tr/+} mice constructed in this paper are considered to be behaviorally consistent with the PTHS syndrome model only by performing the absentee field experiment, and it is suggested to add other relevant behavioral experiments, such as the Marble Burying Test and the Ultrasonic Vocalization Test.

We would like to emphasize that the *Tcf4^{tr/+}* mouse model used here is an accepted model of PTHS that has been shown to exhibit several PTHS associated behaviors including increased repetitive behavior (grooming) and diminished ultrasonic vocalization (PMID: 27568567). For purposes of functional validation, we confirmed the *Tcf4^{tr/+}* mouse used here share some common behavioral abnormalities with other mouse models of this disease including hyperactivity and decreased anxiety (new Fig. 1). However, we do not consider it necessary to show in our hands that *Tcf4^{tr/+}* mice recapitulate all PTHS associated behaviors.

2. The legends in Fig.3A and B are not clearly labeled and confusing. Fig. 3E-I is not mentioned accordingly in the manuscript.

Sorry for this oversight. We have reviewed the text to confirm that all data including the figure panels in question (now presented in Fig. 5) are discussed in the text.

3. Many elaborations in the manuscript are not clear enough. PF-04531083 is mentioned first in Supplementary Figure 4A without any explanation of its role, and in the last part of manuscript that PF-04531083 is mentioned as a blocker of Nav1.8. The order of the supplementary material needs to be better organized.

Thank you for bringing this to our attention. We have made extensive edits to the text and reorganized figures to improve clarity and flow.

4. The detection of reduced CO₂/H⁺ sensitivity at the synaptic level of chemosensitive RTN neurons in slices from TCF4^{tr/+} mice, is synaptic morphology also altered?

Thank you for this query. This is an interesting and likely possibility especially considering RTN neurons in *Tcf4*^{tr/+} mice show morphological changes (clustering) at the cellular level (new Fig. 4). However, we consider analysis of synaptic morphology to be beyond the scope of this study.

5. TCF4 is required for the differentiation of the Atoh1 group of cell subpopulations, and Atoh1 is required for the development of Phox2b-expressing parafacial neurons, single-cell sequencing and other experiments suggest interactions between TCF4 and Atoh1 contribute to cell-type specific deficits in PTHS. The authors should perform ISH and/or IF for TCF4 with Atoh1 in TCF4^{+/+} and TCF4^{tr/+} to verify its expression and effect.

We show by single cell RNA sequencing that Phox2b⁺ neurons in the ventral parafacial region express both *Tcf4* and *Atoh1* (new Fig. 4A). To confirm co-localization of *Tcf4* and *Atoh1* in RTN cell populations, we performed fluorescent in situ hybridization (RNAscope) using tissue sections obtained from 12 day old *Tcf4*^{+/+} mice. We found that 89% of cells had co-localized *Tcf4* and *Atoh1* transcript in the RTN, while 11% of cells only had *Tcf4* transcript (new Fig. 4B). To further support these results, we obtained an enriched population of Phox2b⁺ parafacial neurons (from 22 day old Phox2b::TdT⁺ mice) and subsequent qPCR confirmed the expression of both *Tcf4* and *Atoh1* transcript. Together, these results strongly suggest *Tcf4* and *Atoh1* transcripts are co-expressed by parafacial neurons.

6. Atoh1 affects neonatal respiratory efficacy during RTN neuronal development ((Huang et al,2012, van der Heijden ME et al,2018),) and the authors found that deletion of TCF4 haplogroups also affects neonatal respiratory efficacy, as previously reported (Adriano Flora et al,2007), but how TCF4 interacts with Atoh1 in TCF4^{tr/+} mice to affect the respiratory pattern of mice was not elaborated in this manuscript.

To clarify, the study by Flora et al (PMID: 17878293) did not characterize breathing in *Tcf4* deficient mice. However, that study did show that i) *Tcf4* can heterodimerize with *Atoh1* to form a functional DNA-binding complex, and ii) deletion of *Tcf4* disrupted development of brainstem respiratory centers. The basic helix–loop–helix (BHLH) region of *Tcf4* is required for DNA binding and heteromerization with other transcription factors including *Atoh1*. The BHLH domain is deleted in *Tcf4*^{tr/+} mice, thus rendering the product of that allele non-functional. However, recent evidence suggests this truncated product is expressed (PMID: 26971948) and has a dominant negative effect on the other normal gene product (PMID: 28032012).

7. It is mentioned in the method that adult mice were individually housed on a 12:12 light dark cycle in plastic cages with a running wheel for one week before experimentation. Will having a roller in the cages increase the activity of the mice and will it affect the measurement of

respiratory activity of the mice?

Sorry for this confusion. A running wheel was included in the experimental cages for metabolic analysis and so may have impacted these results. To minimize this issue, we only analyzed metabolic data obtained during periods of rest. Animals did not have access to a running wheel at any other time during this study and respiratory activity was analyzed during periods of relative quiescence so do not think locomotor activity confounded interpretation of our respiratory results.

8. Supplementary Figure 9 is mentioned in the manuscript, but Supplementary Figure 9 was not found.

Sorry for this oversight. We have carefully reorganized all figures to improve clarity and readability.

REVIEWER COMMENTS

Reviewer #1 (Remarks to the Author):

The authors have responded adequately to my comments and I have no further comments.

Reviewer #2 (Remarks to the Author):

I am perfectly satisfied with the significant revisions provided by the authors.

This a great study and a nice work.

I don't have any further comments.

Sincerely yours

Reviewer #3 (Remarks to the Author):

The authors have addressed most of my comments.

Response to reviewer comments

Reviewer #1: The authors have responded adequately to my comments and I have no further comments.

Reviewer #2: I am perfectly satisfied with the significant revisions provided by the authors. This a great study and a nice work. I don't have any further comments.

Reviewer #3: The authors have addressed most of my comments.

We thank the reviewers for their many helpful suggestions!